# Softsign: Smooth Sign in Your Optimizer For Better Parameter Heterogeneity Handling

**Dmitrii Feoktistov** [* 1 2 3] **Timofey Belinsky** [* 4 2] **Andrey Veprikov** [* 4 2 5] **Amir Zainullin** [* 4 2]
**Aleksandr Beznosikov** [4 2 6]

## Abstract

Sign-based and LMO-inspired optimizers have recently attracted substantial attention in deep learning due to their strong performance and low memory footprint. However, their fixed-magnitude updates can hurt terminal convergence: they decouple update mechanisms from gradient magnitudes and fail to account for parameter heterogeneity, often leading to oscillation rather than convergence. We propose `SoftSignum`, a smooth relaxation of sign-based optimization that replaces the hard sign map with a temperature-controlled soft-sign transformation, enabling a parameter-wise transition from sign-like updates to magnitude-sensitive SGD-like steps. We complement it with an adaptive quantile-based temperature schedule and extend the same principle to matrix-valued optimizers, obtaining `SoftMuon`. We also develop a generalized geometry-relaxation framework based on strongly convex regularizers and Fenchel conjugates, proving convergence in stochastic non-convex setting. Experiments on diverse deep learning tasks, including LLM pretraining, show that `SoftSignum` and `SoftMuon` consistently improve over their hard sign-based counterparts and standard `AdamW`.

## 1. Introduction

Deep Learning (DL) models have demonstrated outstanding performance across a wide array of practical applications. The core optimization problem addressed in DL is

$$\min_{\theta \in \mathbb{R}^d} f(\theta), \tag{1}$$

[1]HSE University, Moscow, Russia [2]BRAIn Lab, Moscow, Russia [3]Yandex Research, Moscow, Russia [4]MIRAI, Moscow, Russia [5]SB AI Lab, Moscow, Russia [6]Innopolis University, Innopolis, Russia. Correspondence to: Dmitrii Feoktistov <feoktistovdd@my.msu.ru>.

*Proceedings of the 43^{rd} International Conference on Machine Learning*, Seoul, South Korea. PMLR 306, 2026. Copyright 2026 by the author(s).

where $\theta$ denotes the parameters of the neural network and $f$ is the training objective. We work in the stochastic first-order setting: at each iteration $k$ the optimizer does the full gradient $\nabla f(\theta_k)$ directly, but only a stochastic gradient $g_k$ with $\mathbb{E}[g_k] = \nabla f(\theta_k)$, typically evaluated on a mini-batch of training data.

Solving (1) at the scale of modern deep networks is dominated by the choice of optimizer, which controls both sample efficiency and the quality of the final model. A large part of this design space can be derived from a single principle. At iteration $k$, consider a local quadratic model of the loss around the current iterate $\theta_k$: $f(\theta_k + d) \approx f(\theta_k) + \langle g_k, d \rangle + \frac{1}{2} d^\top \nabla^2 f(\theta_k) \, d$, where $g_k$ is a stochastic estimate of $\nabla f(\theta_k)$. Upper bounding the curvature term by a squared norm, $d^\top \nabla^2 f(\theta_k) \, d \leq \frac{1}{\delta} \|d\|^2$, and minimizing the resulting surrogate yields the Steepest Descent (SD) step under $\| \cdot \|$:

$$
\begin{aligned}
d_k^{\text{SD}} &= \arg \min_{d \in \mathbb{R}^d} \left\{ \langle g_k, d \rangle + \frac{1}{2\delta} \|d\|^2 \right\} \\
&= -\delta \, \|g_k\|_* \cdot \arg \min_{\|t\|=1} \langle g_k, t \rangle,
\end{aligned}
\tag{2}
$$

where $\| \cdot \|_*$ is the dual norm (Bernstein & Newhouse, 2024; 2025). Different choices of $\| \cdot \|$ recover established methods. The Euclidean norm yields plain Stochastic Gradient Descent (`SGD`) (Robbins & Monro, 1951). A norm induced by a positive-definite matrix $H_k$, i.e. $\|d\|_{H_k}^2 = d^\top H_k d$, gives quasi-Newton methods with updates of the form $d_k = -\delta (H_k)^{-1} g_k$ (Byrd et al., 2016), and adaptive methods such as `AdaGrad` (Duchi et al., 2011) or `Adam` (Kingma & Ba, 2014; Loshchilov & Hutter, 2019) when $H_k$ is a diagonal estimate of curvature built from past gradients.

Recent work argues that the gradient-magnitude factor $\|g_k\|_*$ in (2) is not essential for deep learning and can be dropped (Bernstein & Newhouse, 2024; Pethick et al., 2025). What remains is a pure Linear Minimization Oracle (LMO) step:

$$d_k^{\text{LMO}} = -\delta \arg \min_{\|t\| \leq 1} \langle g_k, t \rangle. \tag{3}$$

Choosing $\| \cdot \|$ as the $\ell_\infty$ norm reduces (3) to a coordinate-wise sign step and recovers `normSGD` (Hazan et al., 2015), `signSGD` (Bernstein et al., 2018) and `Lion` (Chen et al.,

2023). These methods are competitive on LLM pretraining and robust to heavy-tailed gradient noise (Kornilov et al., 2025; Yadav et al., 2025). In the matrix case, choosing $\|\cdot\|$ as the spectral norm turns (3) into the orthogonalization step of Muon (Jordan et al., 2024; Liu et al., 2025a; Pethick et al., 2025), one of the strongest current optimizers in Deep Learning (Semenov et al., 2025; Wen et al., 2025; Liu et al., 2025a; Takehi et al., 2025; McGinnis et al., 2025).

Despite these empirical successes, LMO-based methods share a structural limitation that hurts the terminal phase of training.

1) **Parameter Heterogeneity.** Many LMO-based optimizers used in deep learning, such as coordinate-wise sign methods and spectral orthogonalization methods, impose fixed-magnitude updates in their respective geometries. This ignores parameter heterogeneity (Mahoney & Martin, 2019; Sagun et al., 2017): in deep networks, parameters have different scales, sensitivities and convergence dynamics.

2) **Constant Magnitude.** By mapping every update to a fixed geometric boundary, via the $\text{sign}(\cdot)$ operator or spectral orthogonalization, LMO-methods decouple the update magnitude from the underlying gradient. A uniform step size is useful early in training, but close to a minimum, where some coordinates have already converged and their gradients have vanished, the LMO operator keeps forcing a full-magnitude step along those directions. This results in oscillation rather than fine-grained convergence.

We argue that effective terminal convergence benefits from a dynamic optimization geometry rather than a rigid one. Parameters with large, noisy gradients must remain in the robust, bounded LMO regime (3), while parameters with vanishing gradients require the fine-grained, linear resolution of Steepest Descent (2). A naive approach to resolving this dichotomy is a "hard switch": abruptly swapping the optimizer from Signum or Muon to SGD at a predetermined training iteration. However, this heuristic has several limitations:

1) **Temporal Uniformity**. A hard switch assumes all parameters in the network are ready for the SGD phase at the exact same iteration, completely ignoring parameter-wise convergence disparities.

2) **Learning Rate Mismatch**. Because LMO steps and SGD steps exist in fundamentally different magnitude scales, an abrupt switch requires complex learning rate adjustment heuristics.

3) **Momentum Incompatibility**. The momentum buffer accumulated during LMO training may be sub-optimal with the updates of SGD.

In this work, we argue that the transition from LMO-based updates to unconstrained gradient descent should not be

heuristic, but rather a mathematically principled dynamic geometry relaxation. To achieve this, we replace the norm-induced quadratic term in (2) by a general regularizer $V$ (see details in Section 3). One can view this as being in the spirit of Mirror Descent (Nemirovskij & Yudin, 1983; Beck & Teboulle, 2003). However, the key difference is that we do not constrain the model parameters $\theta$ to a feasible set. Thus, $V$ does not define the domain of the iterates, but instead shapes the geometry of the update $d$. By annealing a temperature parameter $\tau$, our framework continuously relaxes the constraints on the optimization step.

As a practical instantiation of this framework, we introduce SoftSignum. By selecting a specific coordinate-wise entropic regularizer, the exact closed-form solution to our geometry-relaxed update yields a temperature-controlled hyperbolic tangent transformation. We pair this with a quantile-based temperature schedule that adaptively models the momentum distribution to ensure a smooth, parameter-aware transition. Furthermore, we extend our approach to matrix-based optimizers and derive SoftMuon.

Our main contributions are summarized as follows:

1) **Novel Optimizers**. We derive SoftSignum and SoftMuon, applying parametrized hyperbolic tangent steps to scalar and matrix regimes. Coupled with our proposed quantile-based temperature scheduling, these methods adaptively transition weights based on their individual convergence dynamics.

2) **Generalized Framework for Geometry Relaxation**. We introduce a unified theoretical framework that formalizes the continuous interpolation between strictly bounded norm-constrained optimization and unconstrained descent via a tunable Bregman-like regularizer.

3) **Rigorous Convergence Theory**. We provide a general convergence analysis for this family of optimizers in the stochastic non-convex setting. By uniquely leveraging the Fenchel conjugate of the regularizer, we establish convergence rates and demonstrate how to theoretically measure progress.

4) **Empirical Evaluation**. We first evaluate our methods on next-character prediction, graph neural network training and Large Language Model pretraining tasks. Then we provide analysis of our methods: analyse method robustness to hyperparameters, show how geometry relaxation affects convergence through a local smoothness analysis, investigate how the tail heaviness of the data distribution affects performance. Our code is available at https://github.com/brain-lab-research/softsign.

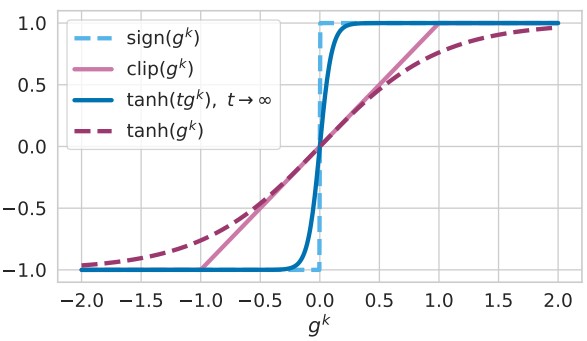

*Figure 1.* Comparison of update transformations. The hard sign map produces constant-magnitude updates for all non-zero inputs, while clipping preserves linear behaviour only near the origin and saturates outside a fixed threshold. The temperature-controlled soft-sign map $\tanh(\tau x)$ smoothly interpolates between these regimes: or large $\tau$ it approaches $\text{sign}(x)$, whereas around the origin it remains linear.

## 2. Smooth Relaxation of Sign-Based Updates

### 2.1. Softsign update

Sign-based optimizers apply a discontinuous map to the gradient or momentum direction. For example, `Signum` updates parameters using $\text{sign}(m_k)$, where $m_k$ is the momentum buffer. Our goal is to replace the hard sign map with a smooth transformation that preserves sign-like behaviour for large momentum components while recovering a linear, `SGD`-like update for small components. We achieve this using a temperature-controlled hyperbolic tangent map $s_\tau(m) = \tanh(\tau m)$ applied coordinate-wise. The temperature parameter $\tau > 0$ controls the optimization geometry. In the high-temperature limit,

$$\tanh(\tau m_i) \to \text{sign}(m_i) \qquad \text{as } \tau \to \infty,$$

and the method recovers the `Signum` update. In contrast, when $|\tau m_i| \ll 1$, we have

$$\tanh(\tau m_i) \approx \tau m_i,$$

and the update becomes locally equivalent to momentum `SGD` with an effective learning rate scaled by $\tau$. Thus, decreasing $\tau$ continuously relaxes the hard sign constraint and allows coordinates with small momentum to take magnitude-aware steps. Figure 1 illustrates this interpolation. Unlike the hard sign map, the soft-sign transformation preserves a linear region around the origin, which allows small-momentum coordinates to take fine-grained steps instead of oscillating.

Given a learning rate $\delta > 0$, momentum buffer $m_k$, decoupled weight decay $\lambda \geq 0$, and temperature $\tau_k$, our proposed `SoftSignum` update is

$$\theta_{k+1} = (1 - \delta\lambda)\theta_k - \delta\tanh(\tau_k m_{k+1}). \qquad (4)$$

This update interpolates smoothly between `Signum` and a clipped momentum-`SGD`-like method without resetting the optimizer state or introducing a discontinuous change in the update direction.

### 2.2. Adaptive Temperature Scheduling

A fixed temperature does not provide a controlled transition between the sign-based and linear regimes. We therefore choose $\tau_k$ adaptively. The goal is to maintain a prescribed fraction of coordinates in the saturated, sign-like regime during the transition.

Let $\alpha = k/N \in [0, 1]$ denote the normalized training progress, where $N$ is the total number of iterations. We keep the optimizer in the pure `Signum` regime until a prescribed transition point $\alpha_{\text{sign}} \in [0, 1]$. That is, for $\alpha < \alpha_{\text{sign}}$, we set $\tau_k = +\infty$ and recover the exact `Signum` update. Once the transition starts, we define the transition progress

$$p_k = \frac{\alpha - \alpha_{\text{sign}}}{1 - \alpha_{\text{sign}}} \in [0, 1]. \qquad (5)$$

At progress $p_k$, we choose a momentum magnitude threshold $q_{p_k}$. Coordinates with $|m_{k,i}| \gtrsim q_{p_k}$ should remain close to saturation, whereas coordinates with $|m_{k,i}| \ll q_{p_k}$ should behave approximately linearly.

To enforce this behaviour, we set $\tau_k$ so that the coordinate with magnitude $q_{p_k}$ is mapped to $1 - \varepsilon$, where $\varepsilon \in (0, 1)$ is a small saturation tolerance:

$$\tanh(\tau_k q_{p_k}) = 1 - \varepsilon.$$

Thus,

$$\tau_k = \frac{\text{arctanh}(1 - \varepsilon)}{q_{p_k}}. \qquad (6)$$

As the transition progresses, the quantile threshold changes, causing the temperature to decrease and gradually moving more coordinates from the saturated regime to the linear regime. In practice, we additionally clip the temperature from below:

$$\tau_k \leftarrow \max\{1, \tau_k\}. \qquad (7)$$

This prevents the effective learning rate in the linear regime from becoming too small.

Computing exact quantiles over all momentum coordinates at every iteration can be expensive. We therefore estimate the momentum magnitude distribution only once, at the beginning of the transition. Since gradient and momentum coordinates in deep networks are often heavy-tailed, we use a folded Cauchy approximation parametrized by robust statistics. Specifically, at $\alpha = \alpha_{\text{sign}}$, we compute the median $\mu$ and median absolute deviation $\sigma$ of the momentum values and approximate the magnitude distribution by $|\text{Cauchy}(\mu, \sigma)|$. Subsequent quantiles $q_{p_k}$ are then obtained from this approximate distribution.

The resulting temperature schedule is summarized in Algorithm 1, which support general saturating map $\psi(\cdot)$, which is $\tanh(\cdot)$ for SoftSignum.

---

**Algorithm 1** $\mathcal{T}(\cdot)$: Temperature Schedule
___

1: **Input:** iteration ratio $\alpha$, momentum vector $m$, sign phase ratio $\alpha_{\text{sign}} \in [0, 1]$, small constant $0 < \varepsilon \ll 1$, smooth invertible function $\psi$.

2: **if** $\alpha = \alpha_{\text{sign}}$ **then**
3:     $\mu \leftarrow \text{median}_i \, m_i$
4:     $\sigma \leftarrow \text{median}_i \, |m_i - \mu|$
5:     Store $\mu, \sigma$ for consequent calls
6: **end if**
7: **if** $\alpha < \alpha_{\text{sign}}$ **then**
8:     $\tau \leftarrow +\infty$
9: **else**
10:    $p \leftarrow \frac{\alpha - \alpha_{\text{sign}}}{1 - \alpha_{\text{sign}}}$
11:    $q_\alpha \leftarrow \text{quantile}\,(p, \mu, \sigma)$
12:    $\tau \leftarrow \dfrac{\psi^{-1}(1 - \varepsilon)}{q_\alpha}$
13:    $\tau \leftarrow \max(1, \tau)$
14: **end if**
15: **Return:** $\tau$

---

### 2.3. SoftSignum

Combining the soft-sign update (4) with the quantile-based temperature schedule gives SoftSignum, presented in Algorithm 2. Before the transition point, SoftSignum coincides with Signum. After the transition starts, the temperature decreases smoothly, and coordinates enter the linear regime depending on their own momentum magnitudes.

This mechanism avoids the main drawbacks of a hard switch from Signum to SGD. First, the transition is parameter-wise: different coordinates move from the sign-like regime to the linear regime at different times. Second, the effective learning rate changes continuously through $\tau_k$, avoiding abrupt scale mismatch between Signum and SGD updates. Third, the momentum buffer is preserved throughout training, and the optimizer retains the directional information accumulated during the sign phase.

### 2.4. Spectral Extension: SoftMuon

The same relaxation principle can be applied to matrix-valued LMO-based optimizers. In particular, Muon can be interpreted as applying a sign operation in the spectral domain. Let $M \in \mathbb{R}^{m \times n}$ be a momentum matrix with singular value decomposition

$$M = U \, \text{diag}(\sigma) V^\top.$$

The Muon direction is

$$\text{Muon}(M) = UV^\top,$$

---

**Algorithm 2** SoftSignum
___

**Require:** Learning rate $\delta > 0$; momentum coefficient $\beta \in (0, 1)$; weight decay $\lambda \geq 0$; number of iterations $N$; transition point $\alpha_{\text{sign}}$; saturation tolerance $\varepsilon$; initial point $\theta_0$.

1: Initialize $m_0 \leftarrow 0$
2: **for** $k = 0, \dots, N - 1$ **do**
3:     Obtain stochastic gradient $g_k$ at $\theta_k$
4:     $m_{k+1} \leftarrow \beta m_k + (1 - \beta) g_k$
5:     $\alpha \leftarrow k/N$
6:     $\tau_k \leftarrow T(\alpha, m_{k+1}, \alpha_{\text{sign}}, \varepsilon, \tanh(\cdot))$
7:     $\theta_{k+1} \leftarrow (1 - \delta\lambda)\theta_k - \delta \tanh(\tau_k m_{k+1})$
8: **end for**

---

which is equivalent to replacing every non-zero singular value by one. Therefore, Muon inherits the same terminal-phase limitation as coordinate-wise sign methods: small singular directions still receive a full-magnitude update.

To obtain a smooth spectral relaxation, we replace the hard sign operation on singular values by a smooth saturating function. Although one can use $\tanh(\tau\sigma_i)$ directly, in the matrix case it is computationally convenient to use the algebraic soft-sign function

$$\phi(x) = \frac{x}{\sqrt{1 + x^2}}.$$

Applied to a momentum matrix, this gives the spectral map, which can be efficiently computed using Newton–Schulz iterations (Jiang et al., 2026)

$$\Phi_\tau(M) = U \, \text{diag}\left(\frac{\tau\sigma_i}{\sqrt{1 + \tau^2\sigma_i^2}}\right) V^\top.$$

Equivalently,

$$\Phi_\tau(M) = M \left(M^\top M + \tau^{-2}I\right)^{-1/2}. \tag{8}$$

For large $\tau$, this recovers the Muon direction:

$$\Phi_\tau(M) \to UV^\top.$$

For small singular values satisfying $\tau\sigma_i \ll 1$, the map is approximately linear:

$$\frac{\tau\sigma_i}{\sqrt{1 + \tau^2\sigma_i^2}} \approx \tau\sigma_i.$$

Thus, SoftMuon provides the same type of smooth transition as SoftSignum, but in the spectral geometry used by Muon.

The temperature schedule for SoftMuon follows the same quantile-based principle as in SoftSignum, but operates on the singular values of the momentum matrix. At the transition point, we compute the singular values of $M_{\alpha_{\text{sign}}}$ and

---

**Algorithm 3** `SoftMuon`

---

**Require:** Learning rate $\delta > 0$; momentum coefficient $\beta \in (0, 1)$; weight decay $\lambda \geq 0$; number of iterations $N$; transition point $\alpha_{\text{sign}}$; saturation tolerance $\varepsilon$; initial point $\Theta_0$.

1: Initialize $M_0 \leftarrow 0$
2: **for** $k = 0, \ldots, N - 1$ **do**
3:     Obtain stochastic gradient $G_k$ at $\Theta_k$
4:     $M_{k+1} \leftarrow \beta M_k + (1 - \beta) G_k$
5:     $\alpha \leftarrow k/N$
6:     **if** $\alpha = \alpha_{\text{sign}}$ **then**
7:         Compute singular values $\sigma_{\alpha_{\text{sign}}}$ of $M_{k+1}$
8:     **end if**
9:     $\tau_k \leftarrow T(\alpha, \sigma_{\alpha_{\text{sign}}}, \alpha_{\text{sign}}, \varepsilon, \phi(\cdot))$
10:    $\Delta \Theta_k \leftarrow M_{k+1} \left( M_{k+1}^\top M_{k+1} + \tau_k^{-2} I \right)^{-1/2}$
11:    $\Theta_{k+1} \leftarrow (1 - \delta \lambda) \Theta_k - \delta \Delta \Theta_k$
12: **end for**

---

estimate their magnitude distribution. During the transition, the temperature is chosen so that a prescribed quantile of singular values remains close to saturation. The resulting algorithm is shown in Algorithm 3.

### 2.5. Practical Defaults

`SoftSignum` introduces only a small number of additional hyperparameters: the transition point $\alpha_{\text{sign}}$, the saturation tolerance $\varepsilon$, and the number of Newton iterations $N_q$ used for quantile computation. Unless stated otherwise, we use

$$\alpha_{\text{sign}} = 0.9, \qquad \varepsilon = 10^{-4}, \qquad N_q = 10.$$

These values provide a simple default configuration and allow `SoftSignum` to be integrated into existing `Signum` pipelines without additional tuning. The same defaults are used for `SoftMuon` unless specified otherwise in the corresponding experiments.

## 3. Generalized Framework and Convergence

### 3.1. Generalized Update

The update rule of `SoftSignum` (Algorithm 2) can be expressed in a more general form. Consider the following parametrized update at iteration $k$:

$$\theta_{k+1} = \theta_k + \delta \arg \min_{d \in \mathcal{D}} \left\{ \langle m_k, d \rangle + \frac{1}{\tau} V(d) \right\}, \quad (9)$$

where $\delta > 0$ is the learning rate, $\tau > 0$ is the temperature parameter, $m_k$ is the momentum buffer, and $V : \mathbb{R}^d \to \mathbb{R}_+$ is a regularization function that controls the geometry of the update step. This formulation encompasses a broad family of optimization methods, including standard `SGD`,

sign-based methods, and our proposed `SoftSignum`, by appropriate choice of $V$.

**(i) Euclidean norm:** Taking $V(u) = \frac{1}{2} \|u\|_2^2$ with $\mathcal{D} = \mathbb{R}^d$ recovers standard momentum `SGD` with the update $\theta_{k+1} = \theta_k - \delta \tau m_k$.

**(ii) General norms:** For an arbitrary norm $\|\cdot\|$, one can take $V(u) = \frac{1}{2} \|u\|^2$ and $\mathcal{D} = \mathbb{R}^d$. This yields norm-adapted methods like `Muon` (Jordan et al., 2024) or `Lion` (Chen et al., 2023).

**(iii) `SoftSignum`:** For $\mathcal{D} = (-1, 1)^d$,

$$V(u) = \frac{1}{2} \sum_{i=1}^{d} \left[ (1 + u_i) \ln(1 + u_i) + (1 - u_i) \ln(1 - u_i) \right],$$

the corresponding update is $\theta_{k+1} = \theta_k - \delta \tanh(\tau m_k)$, where $\tanh$ is applied element-wise. This is precisely the update rule employed by our `SoftSignum` method (Algorithm 2 with fixed temperature and without weight decay).

**(iv) Algebraic `SoftSignum`:** For $\mathcal{D} = (-1, 1)^d$,

$$V(u) = \sum_{i=1}^{d} \left( 1 - \sqrt{1 - u_i^2} \right),$$

the corresponding update is

$$\theta_{k+1} = \theta_k - \delta \cdot \frac{\tau m_k}{\sqrt{1 + \tau^2 m_k^2}},$$

where the operation is applied coordinate-wise. The function $u/\sqrt{1 + u^2}$ is an algebraic approximation to $\tanh(u)$ that avoids transcendental operations. This is particularly useful for matrix-valued updates, where applying the step to the singular values of a gradient matrix can be implemented via Newton–Schulz iterations (Jiang et al., 2026).

The derivation of the updates **(iii)** and **(iv)** is provided in Appendix A.

### 3.2. Convergence Analysis

We focus on the non-convex stochastic setting standard in theoretical deep learning (Ghadimi & Lan, 2013; Reddi et al., 2019; Yan et al., 2018). Incorporating momentum gives variance reduction at the cost of coupling iterates across time, and arbitrary regularizers $V$, in particular the non-quadratic $V$ used by `SoftSignum`, preclude a norm-based proof.

We impose the following structural assumptions on the regularizer $V$.

**Assumption 3.1** (Properties of $V$). $V : \mathcal{D} \to \mathbb{R}$ is proper, closed and convex on a convex set $\mathcal{D} \subseteq \mathbb{R}^d$ with $0 \in \mathcal{D}$ and $V(0) = 0$, and is 1-strongly convex with respect to the Euclidean norm: for all $d, d' \in \mathcal{D}$,

$$V(d') \geq V(d) + \langle \nabla V(d), d' - d \rangle + \frac{1}{2} \|d' - d\|_2^2. \quad (10)$$

Assumption 3.1 holds for the Euclidean case $V(d) = \frac{1}{2}\|d\|_2^2$ and for both `SoftSignum` regularizers (Appendix A). A norm-dependent variant under 1-strong convexity with respect to an arbitrary $\|\cdot\|$ is treated in Appendix C. We further use the standard Euclidean assumptions (Nesterov, 2004; Nemirovski et al., 2009).

**Assumption 3.2** (Smoothness). The objective function $f : \mathbb{R}^d \to \mathbb{R}$ is $L$-smooth, i.e., for all $\theta, \theta' \in \mathbb{R}^d$,

$$\|\nabla f(\theta) - \nabla f(\theta')\|_2 \leq L\|\theta - \theta'\|_2.$$

**Assumption 3.3** (Stochastic gradients). Let $\{\mathcal{F}_k\}_{k \geq 0}$ be a filtration such that $\theta_k$ is $\mathcal{F}_k$-measurable. At each iteration $k$ we have access to a stochastic gradient $g_k$ satisfying

$$\mathbb{E}[g_k \mid \mathcal{F}_k] = \nabla f(\theta_k), \qquad \mathbb{E}[\|g_k - \nabla f(\theta_k)\|_2^2 \mid \mathcal{F}_k] \leq \sigma^2,$$

where $\sigma^2 \geq 0$ is the variance bound.

The main convergence guarantee is the following.

**Theorem 3.4** (Convergence rate). *Let Assumptions 3.1, 3.2, and 3.3 hold. Consider the momentum method with updates*

$$m_k = \beta m_{k-1} + (1 - \beta)g_k,$$
$$\theta_{k+1} = \theta_k + \delta \arg\min_{d \in \mathcal{D}} \left\{ \langle m_k, d \rangle + \frac{1}{\tau} V(d) \right\}.$$

*Let $\beta \in (0, 1)$ and suppose*

$$\delta \cdot \tau \leq \frac{1}{2L} \cdot \min\left\{ 1 \; ; \; \frac{1 - \beta}{\beta} \right\}.$$

*Then for any $K \geq 1$,*

$$\frac{1}{K} \sum_{k=0}^{K-1} \frac{1}{\tau^2} \mathbb{E}[V^*(-\tau \nabla f(\theta_k))]$$
$$\leq \frac{f(\theta_0) - f^*}{\delta \tau K} + (1 - \beta)\sigma^2 + \frac{\|\nabla f(\theta_0)\|_2^2}{K(1 - \beta)},$$

*where $V^*(y) = \sup_{u \in \mathcal{D}}\{\langle y, u \rangle - V(u)\}$ is the Fenchel conjugate of $V$, and $f^* := \inf_{\theta \in \mathbb{R}^d} f(\theta) > -\infty$.*

*Proof.* See Appendix B. $\square$

**Corollary 3.5** (Iteration complexity). *Under the assumptions of Theorem 3.4, choose*

$$\beta = 1 - \min\left\{ 1 \; ; \; \sqrt{\frac{2L(f(\theta_0) - f^*) + \|\nabla f(\theta_0)\|_2^2}{K\sigma^2}} \right\},$$

$$\delta \cdot \tau = \frac{1}{2L} \cdot \min\left\{ 1 \; ; \; \frac{1 - \beta}{\beta} \right\}.$$

*Then to ensure*

$$\frac{1}{K} \sum_{k=0}^{K-1} \frac{1}{\tau^2} \mathbb{E}[V^*(-\tau \nabla f(\theta_k))] \leq \varepsilon^2,$$

*it suffices to make*

$$K = \mathcal{O}\left( \left[ L(f(\theta_0) - f^*) + \|\nabla f(\theta_0)\|_2^2 \right] \cdot \max\left\{ \frac{1}{\varepsilon^2} ; \frac{\sigma^2}{\varepsilon^4} \right\} \right)$$

*iterations of Algorithm 2.*

### 3.3. Discussion of the Convergence Rate

**Interpretation of $V^*$.** Theorem 3.4 measures progress through the Fenchel conjugate $V^*(-\tau \nabla f(\theta_k))$, which generalizes the squared gradient norm to the dual geometry induced by $V$. For the Euclidean choice $V(u) = \frac{1}{2}\|u\|_2^2$ we have $V^*(y) = \frac{1}{2}\|y\|_2^2$, the criterion in Theorem 3.4 reduces to the classical $\frac{1}{K} \sum_k \mathbb{E}\|\nabla f(\theta_k)\|_2^2$, and Corollary 3.5 recovers the standard non-convex stochastic momentum rate (Yan et al., 2018).

**SoftSignum case.** For the coordinate-wise `SoftSignum` regularizer,

$$V^*(y) = \sum_{i=1}^{d} \ln[\cosh(y_i)].$$

This quantity behaves quadratically for small gradients and linearly for large ones. If gradients are large for $k < K_1$ and small afterwards, and using the asymptotics $\frac{1}{\tau^2} \ln[\cosh(\tau x)] \approx \frac{|x|}{\tau}$ for large $x$ and $\frac{1}{\tau^2} \ln[\cosh(\tau x)] \approx \frac{x^2}{2}$ for small $x$, Corollary 3.5 approximately becomes

$$\frac{1}{K} \left( \frac{1}{\tau} \sum_{k < K_1} \mathbb{E}[\|\nabla f(\theta_k)\|_1] + \sum_{k \geq K_1} \mathbb{E}[\|\nabla f(\theta_k)\|_2^2] \right) \lesssim \varepsilon^2.$$

Compared to the Euclidean criterion $\frac{1}{K} \sum_{k=0}^{K-1} \mathbb{E}[\|\nabla f(\theta_k)\|_2^2] \leq \varepsilon^2$, `SoftSignum` measures progress via the $\ell_1$ norm during the large-gradient regime. Since $\|\cdot\|_1 \geq \|\cdot\|_2$ and the coefficient $1/\tau$ can be larger than 1, `SoftSignum` enforces a stricter early-stage convergence criterion. Despite this, the iteration complexity matches the Euclidean case.

**Temperature interpretation.** Our analysis reveals that $\tau$ plays a role symmetric to $\delta$ in controlling convergence: the effective step size is $\tau\delta$. This provides theoretical insight into why $\tau$ scheduling is desirable: it is well-established that learning rate scheduling is profitable for training neural networks, and the key advantage of temperature formulation is that it allows for parameter-wise learning rate scheduling.

### 3.4. Matrix Extensions

The framework of Section 3 extends naturally to matrix-valued parameters. For any scalar regularizer $V$ satisfying Assumption 3.1, define the spectral function

$$\mathbb{V}(D) = \sum_i V(\sigma_i(D)), \tag{11}$$

where $D \in \mathbb{R}^{m \times n}$ and $\sigma_i(D)$ are its singular values. The matrix update replaces the vector update (9) by

$$\Theta_{k+1} = \Theta_k + \delta \arg \min_{D \in \mathcal{D}} \left\{ \langle M_k, D \rangle_F + \frac{1}{\tau} \mathbb{V}(D) \right\}, \quad (12)$$

where $\langle \cdot, \cdot \rangle_F$ is the Frobenius inner product and $M_k$ is the matrix-valued momentum.

The convergence proof in Appendix B uses the Fenchel conjugate of $V$ through the duality identity. By Lewis (1996) (Theorem 2.3, eq. (1.3)), the Fenchel conjugate of $\mathbb{V}$ is again a spectral function:

$$\mathbb{V}^*(Y) = \sum_i V^*(\sigma_i(Y)). \quad (13)$$

This means every step of the scalar proof carries over to the matrix case with $\|\cdot\|_2$ replaced by $\|\cdot\|_F$ in Assumptions 3.2 and 3.3.

The closed-form solution to (12) follows from Lewis (1996) (Corollary 3.2). For the algebraic regularizer (Example (iv)), let $M_k = \hat{U} \operatorname{diag}(\hat{\sigma}) \hat{V}^\top$ be the SVD of the momentum matrix. Then

$$\Theta_{k+1} = \Theta_k - \delta \hat{U} \operatorname{diag}\left( \frac{\tau \hat{\sigma}}{\sqrt{1 + \tau^2 \hat{\sigma}^2}} \right) \hat{V}^\top, \quad (14)$$

where all operations on $\hat{\sigma}$ are coordinate-wise. This update can be computed without an explicit SVD via Newton–Schulz iterations (Jiang et al., 2026). For the algebraic regularizer, $V^*(y) = \sqrt{1 + y^2} - 1$, therefore (13) gives $\mathbb{V}^*(Y) = \sum_i \left( \sqrt{1 + \sigma_i(Y)^2} - 1 \right)$, which is a smooth approximation to the nuclear norm.

**Theorem 3.6** (Matrix convergence rate)**.** *Under the conditions of Theorem 3.4 with $\|\cdot\|_2$ replaced by $\|\cdot\|_F$ in Assumptions 3.2 and 3.3, and $V$ replaced by $\mathbb{V}$ in the update (12), the same step condition holds and for any $K \geq 1$ it holds that:*

$$\frac{1}{K} \sum_{k=0}^{K-1} \frac{1}{\tau^2} \mathbb{E}[\mathbb{V}^*(-\tau \nabla f(\Theta_k))] \leq \frac{f(\Theta_0) - f^*}{\delta \tau K}$$

$$+ (1 - \beta)\sigma^2 + \frac{\|\nabla f(\Theta_0)\|_F^2}{K(1 - \beta)},$$

*where $\mathbb{V}^*$ is given by (13).*

*Proof.* The proof is identical to Appendix B, replacing $\langle \cdot, \cdot \rangle$ with $\langle \cdot, \cdot \rangle_F$, $\|\cdot\|_2$ with $\|\cdot\|_F$, and $V^*$ with $\mathbb{V}^*$ via (13). □

The convergence rate, step-size condition, and dependence on $L$, $\sigma^2$, and $\beta$ are identical to the scalar case. The identity (13) is the only structural property of $V^*$ used in the proof, and any future refinement of Theorem 3.4 therefore transfers to the matrix setting at no additional cost.

## 4. Experiments

### 4.1. Next-character prediction

We first evaluate on a small-scale next character prediction (NCP) task built from the multilingual NLI corpus (Laurer et al., 2022), using English, Spanish, and Italian texts. We train a character-level LSTM (Hochreiter & Schmidhuber, 1997) and a causal character-level Transformer (Vaswani et al., 2017); details are in Appendix E.1. We compare against standard baselines and `HardSwitchSign`, which runs `Signum` for the first $\alpha_{\text{sign}}$ fraction of training and then switches to momentum `SGD`. Table 1 shows that `SoftSignum` is the best coordinate-wise optimizer on both architectures, outperforming `AdamW`, `Signum`, `SGD`, and `HardSwitchSign`. On the Transformer, `SoftMuon` further improves over `Muon` and achieves the best overall accuracy.

*Table 1.* Character-level accuracy (%) on next-character prediction. Results are averaged over 5 runs; margins correspond to 95% confidence intervals.

| Method | Transformer | LSTM |
|---|---|---|
| SoftMuon | **57.84 ± 0.04** | **63.02 ± 0.07** |
| Muon | 57.36 ± 0.08 | 63.02 ± 0.07 |
| SoftSignum | **57.05 ± 0.12** | **61.36 ± 0.12** |
| AdamW | 56.91 ± 0.14 | 60.07 ± 0.12 |
| HardSwitchSign | 56.17 ± 0.08 | 60.40 ± 0.16 |
| Signum | 56.17 ± 0.11 | 59.71 ± 0.13 |
| SGD | 48.96 ± 0.38 | 31.40 ± 0.19 |
| SGD + LinearLR | 49.22 ± 0.67 | 27.89 ± 0.13 |

### 4.2. GNN training

*Table 2.* Graph Transformer results on GraphLand datasets. Results are averaged over 8 runs and reported as mean ± standard deviation. Bold denotes the best mean performance for each dataset. Underlining denotes methods statistically tied with the best result within one standard deviation. AR denotes average rank across datasets, and Margin denotes the average performance gap to the best optimizer on each dataset.

| | SoftMuon | Muon | SoftSignum | Signum | AdamW |
|---|---|---|---|---|---|
| AR | 1.86 | 2.43 | 2.71 | 3.86 | 4.14 |
| Margin | 0.21 | 0.84 | 1.25 | 1.62 | 2.61 |
| tolokers-2 (ROC-AUC ↑) | 56.21 ± 0.48 | 56.18 ± 0.65 | **57.23 ± 0.34** | 55.85 ± 0.59 | 56.17 ± 0.41 |
| artnet-views ($R^2$ ↑) | **55.31 ± 0.45** | 54.09 ± 0.58 | 54.05 ± 0.91 | 52.14 ± 0.41 | 53.78 ± 0.41 |
| hm-prices ($R^2$ ↑) | **71.10 ± 0.46** | 68.89 ± 1.20 | 71.06 ± 0.76 | 70.43 ± 0.64 | 62.31 ± 2.05 |
| city-roads-M ($R^2$ ↑) | **59.54 ± 0.13** | 59.40 ± 0.14 | 58.53 ± 0.54 | 58.44 ± 0.54 | 55.93 ± 0.53 |
| artnet-exp ($R^2$ ↑) | **44.34 ± 0.31** | 43.11 ± 0.23 | 43.21 ± 0.44 | 43.42 ± 0.30 | 42.90 ± 0.32 |
| avazu-ctr ($R^2$ ↑) | 30.56 ± 0.79 | **30.73 ± 0.43** | 25.47 ± 1.86 | 26.85 ± 1.85 | 28.97 ± 0.53 |
| city-reviews (ROC-AUC ↑) | 77.89 ± 0.11 | **78.17 ± 0.11** | 78.09 ± 0.09 | 77.93 ± 0.12 | 78.07 ± 0.08 |

We next evaluate `SoftSignum` and `SoftMuon` on graph learning tasks. We use the GraphLand benchmark (Bazhenov et al., 2025), which contains diverse real-world industrial graph datasets, and train a Graph Transformer backbone (Shi et al., 2020). We compare against `Signum`, `Muon`, and `AdamW` under the same training protocol. Results are averaged over 8 random seeds and reported in Table 2. `SoftMuon` achieves the best aggregate performance, obtaining the lowest average rank and the smallest average

margin to the best optimizer across datasets. `SoftSignum` also improves over its hard-sign counterpart `Signum` in both average rank and average margin. Full architecture, training, and hyperparameter-tuning details are provided in Appendix E.2

### 4.3. LLM pretraining

We evaluate the proposed methods in LLM pretraining. First, we train a 130M-parameter LLaMA-based (Touvron et al., 2023) model on C4 (Raffel et al., 2020) following the protocol of Zmushko et al. (2024). We compare `SoftSignum` against `Signum` and evaluate `SoftMuon` against `Muon` and `AdamW`. Second, we train a 360M-parameter SmolLM2-style (Allal et al., 2025) model on FineWeb-Edu (Penedo et al., 2024) and compare `SoftMuon`, `Muon`, and `AdamW`. Figure 2 reports evaluation perplexity around the transition point. On the 130M model, `SoftSignum` improves over `Signum` after the transition starts, indicating that smooth relaxation helps recover magnitude-sensitive terminal convergence while retaining the benefits of sign-based training. `SoftMuon` achieves the lowest evaluation perplexity among the tested optimizers. The same pattern holds in the 360M SmolLM2 experiment, where `SoftMuon` improves over `Muon` after the transition and reaches the best final perplexity. To verify improvements scale further, we also train a 720M model on FineWeb using the setup of Semenov et al. (2025). `SoftMuon` achieves 16.216 perplexity vs. 16.362 for `Muon`. Full training details, parameter-group rules, and hyperparameters are provided in Appendix E.3.

## 5. Method Analysis

### 5.1. Robustness to $\alpha_{sign}$

Our methods introduce one additional important hyperparameter $\alpha_{sign}$. We investigate robustness of `SoftSignum` to this hyperparameter using setups from Section 4.3 and 4.1. As shown in Table 3 `SoftSignum` is highly robust to the choice of $\alpha_{sign}$. When varying this hyperparameter across a range from 0.9 down to 0.3, the final metrics remain stable.

*Table 3.* Robustness of `SoftSignum` to the $\alpha_{sign}$ hyperparameter across different models and tasks.

| $\alpha_{sign}$ | 0.9 | 0.7 | 0.5 | 0.3 |
|---|---|---|---|---|
| Transformer, NCP (val acc) | 57.05 | 57.14 | 57.19 | 57.20 |
| 130M LLaMa, pretraining (perplexity) | 18.519 | 18.530 | 18.541 | 18.559 |

As a default baseline value that allows integrating `SoftSignum` into existing pipelines without additional tuning, we propose using $\alpha_{sign} = 0.9$. A significant practical advantage of this choice is that it enables the efficient

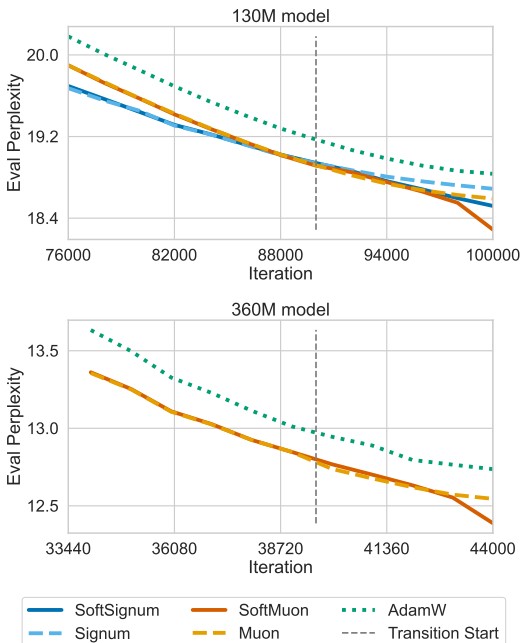

*Figure 2.* Training progress of different optimization methods during 130M LLaMa-based architecture pretraining on C4 dataset (upper) and 360M SmolLM2 on FineWeb-Edu (lower).

reuse of existing model checkpoints trained with a standard baseline (e.g., `Signum` or `Muon`).

### 5.2. Unbalanced CIFAR

Sign-based methods are known to be beneficial under heavy-tailed data distribution (Yadav et al., 2025; Kornilov et al., 2025). To investigate `SoftSignum` performance under different data distribution, we merge the original CIFAR-10 classes (Krizhevsky et al., 2009) into two superclasses by label parity and downsample one superclass in the train, validation, and test splits. The imbalance coefficient is $k = $ #samples in superclass $0$/#samples in superclass 1, with $k \in \{1, 2, 5, 10, 30, 40, 50, 60, 80, 100, 200\}$. As model we use simple CNN (see Appendix E.4).

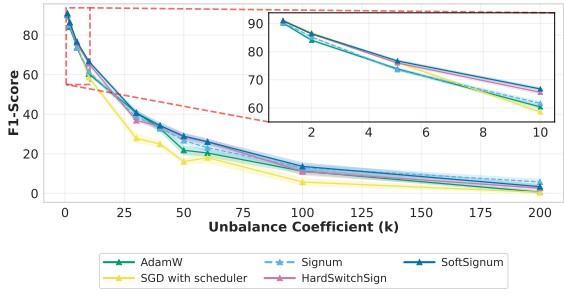

*Figure 3.* Optimizer comparison on imbalanced CIFAR-10. Final F1-score is averaged over 40 runs.

Figure 3 shows that scheduled `SGD` performs best, or statistically comparably to the best method, under mild imbalance ($k \leq 5$), consistent with its strong performance on homogeneous vision tasks (Zhang et al., 2020). However, its performance drops rapidly for $k \geq 10$. In this regime, `SoftSignum` outperforms `SGD` and `AdamW`, while pure `Signum` becomes strongest under extreme imbalance ($k \geq 100$). Thus, the intermediate imbalance regime is where the smooth transition between sign-based and magnitude-sensitive updates is most useful. Additional details can be found in Appendix E.4. We extend this type of analysis to Softmax Unigram Model (Yadav et al., 2025) in Appendix D.

### 5.3. Smoothness Analysis

We further examine how the optimization geometry changes during the `SoftSignum` transition. Prior work has shown that sharpness and local curvature can vary substantially along the training trajectory, and that such changes are informative for understanding optimizer dynamics and stability (Wang et al., 2022; Riabinin et al., 2025). Following this line of analysis, we estimate the local smoothness at iteration $k$ using the same setup as in Section 4.1:

$$L^k = \frac{\left\| \frac{1}{B} \sum_{j=1}^{B} \left( \nabla f_{i_j}(\theta^k) - \nabla f_{i_j}(\theta^{k-1}) \right) \right\|_2}{\|\theta^k - \theta^{k-1}\|_2}, \quad (15)$$

where $B$ denotes a fixed mini-batch that is kept constant throughout the entire training process. All values $L^k$ within one epoch are averaged and logged. Figure 4 shows that

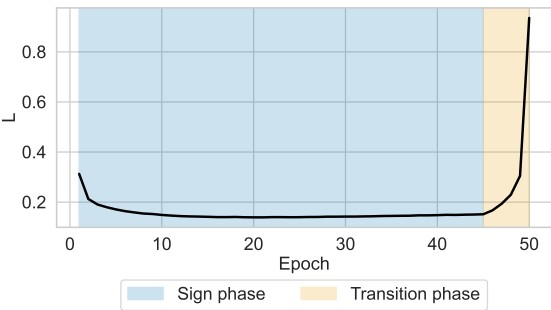

*Figure 4.* Local smoothness estimate during different optimization steps of `SoftSignum`: sign phase, transition phase. See Appendix E.5 for details.

the estimated local smoothness increases after the transition starts. We interpret this as an indication that the update dynamics change during the transition: the sign phase largely normalizes coordinate-wise update magnitudes, while the relaxed phase makes updates more sensitive to gradient scale. This interpretation is consistent with prior analyses connecting trajectory-dependent smoothness and sharpness to optimizer behaviour (Jastrzebski et al., 2019; Wang et al., 2022; Riabinin et al., 2025). Thus, the observed increase

suggests that `SoftSignum` changes the effective optimization regime.

## 6. Related work

**Linear Minimization Oracle.** A classical approach to solving (1) is to employ norm-constrained optimizers, where each step is obtained by querying a LMO over a prescribed unit ball (Pethick et al., 2025), thus connecting modern deep-learning training to conditional-gradient and mirror-type perspectives. Early representatives of this line include `normSGD` (Hazan et al., 2015) and `signSGD` (Bernstein et al., 2018), which can be interpreted as inducing LMO-like directions under particular norm geometries. More recently, this viewpoint has been used to motivate and unify practical optimizers such as `Lion` (Chen et al., 2023) and `Muon` (Jordan et al., 2024). LMO-based training has demonstrated strong empirical performance (Liu et al., 2025a; Team et al., 2025). Beyond the base algorithm, `Muon` has been studied through a number of scalable variants and constraint-aware modifications (Riabinin et al., 2025; Amsel et al., 2025; Liu et al., 2025b; Huang et al., 2025).

**Optimizer switching and clipping-based transitions.** Optimizer switching methods aim to combine the fast initial progress of adaptive methods with the late-stage behaviour of SGD. SWATS (Keskar & Socher, 2017) switches from Adam to SGD once the Adam step can be approximated by a scalar SGD step, while DSTAdam (Zeng et al., 2022) uses a decreasing scaling transition from Adam to SGD to avoid an abrupt change. These approaches, however, apply the transition globally across parameters. Our method is also related to generalized clipping. Pethick et al. (2026) interpret gradient norm clipping as a hybrid between steepest descent and LMO-type updates under generalized $(L_0, L_1)$-smoothness. This perspective is complementary to ours: their framework justifies clipping-like updates through non-Euclidean geometry, whereas we use a smooth saturating map to relax a sign-based geometry into an SGD-like one. The `SoftSignum` update $\tanh(\cdot)$ can be viewed as a smooth coordinate-wise analogue of clipping. Thus, decreasing the temperature $\tau$ implements a gradual, parameter-wise transition from the bounded LMO regime to magnitude-sensitive descent.

## 7. Conclusion

We introduced `SoftSignum` and `SoftMuon`, smooth relaxations of sign-based and spectral LMO-based optimizers that enable parameter-wise transitions from bounded updates to magnitude-sensitive descent. Our generalized framework provides convergence guarantees in stochastic non-convex settings, while experiments across diverse deep learning tasks demonstrate consistent improvements over hard sign-based baselines.

## Acknowledgments

The work was supported by the Ministry of Economic Development of the Russian Federation (agreement No. 139-15-2025-013, dated June 20, 2025, IGK 000000C313925P4B0002).

## Impact Statement

This paper presents work whose goal is to advance the field of Machine Learning. There are many potential societal consequences of our work, none of which we feel must be specifically highlighted here.

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

## A. SoftSignum Regularizer Derivation

In this section, we derive the update rules for the `SoftSignum` regularizers (Examples (iii) and (iv) in the main text) and verify that they satisfy Assumption 3.1.

**Lemma A.1** (Equivalence of update formulations). *The update rule* (9),

$$\theta_{k+1} = \theta_k + \delta \arg\min_{d \in \mathcal{D}} \left\{ \langle m_k, d \rangle + \frac{1}{\tau} V(d) \right\},$$

*is equivalent to*

$$\theta_{k+1} = \theta_k + \arg\min_{d \in \delta \mathcal{D}} \left\{ \langle m_k, d \rangle + \frac{\delta}{\tau} V \left( \frac{d}{\delta} \right) \right\}. \tag{16}$$

*Proof.* Let $d^* = \arg\min_{d \in \mathcal{D}} \{ \langle m_k, d \rangle + \frac{1}{\tau} V(d) \}$. Setting $\tilde{d} = \delta d^*$, we have $\tilde{d} \in \delta \mathcal{D}$ and

$$\langle m_k, \tilde{d} \rangle + \frac{\delta}{\tau} V \left( \frac{\tilde{d}}{\delta} \right) = \delta \left( \langle m_k, d^* \rangle + \frac{1}{\tau} V(d^*) \right).$$

Since multiplying the objective by $\delta > 0$ does not change the minimizer, $\tilde{d}$ minimizes the right-hand side over $\delta \mathcal{D}$. Therefore $\arg\min_{d \in \delta \mathcal{D}} \{ \langle m_k, d \rangle + \frac{\delta}{\tau} V(d/\delta) \} = \delta d^*$, and both formulations produce the same iterate $\theta_{k+1} = \theta_k + \delta d^*$. □

For brevity, the remainder of this paper uses the form (16).

### A.1. Derivation of the Update Rule

Both regularizers are defined coordinate-wise on $\mathcal{D} = (-\delta, \delta)^d$ via $W_k(d) = \frac{\delta}{\tau} V(d/\delta)$. The update step solves $\Delta \theta_k = \arg\min_{d \in \mathcal{D}} \{ \langle m_k, d \rangle + W_k(d) \}$, whose first-order condition for coordinate $i$ is

$$m_{k,i} + \frac{1}{\tau} \frac{\partial V}{\partial u} \left( \frac{d_i}{\delta} \right) = 0. \tag{17}$$

**tanh regularizer** (Example (iii)).

$$V(u) = \sum_{i=1}^{d} \left[ \tfrac{1}{2}(1 + u_i) \ln(1 + u_i) + \tfrac{1}{2}(1 - u_i) \ln(1 - u_i) \right], \qquad \frac{\partial V}{\partial u}(u) = \tfrac{1}{2} \ln \tfrac{1+u}{1-u} = \operatorname{arctanh}(u).$$

The condition (17) gives $\operatorname{arctanh}(d_i/\delta) = -\tau m_{k,i}$, so $d_i = -\delta \tanh(\tau m_{k,i})$, and

$$\theta_{k+1} = \theta_k - \delta \tanh(\tau m_k), \tag{18}$$

where `tanh` is applied element-wise. This is the `SoftSignum` update (Algorithm 2).

**Algebraic regularizer** (Example (iv)).

$$V(u) = \sum_{i=1}^{d} \left( 1 - \sqrt{1 - u_i^2} \right), \qquad \frac{\partial V}{\partial u}(u) = \frac{u}{\sqrt{1 - u^2}}.$$

The condition (17) gives $\frac{d_i/\delta}{\sqrt{1 - d_i^2/\delta^2}} = -\tau m_{k,i}$. Squaring and solving: $d_i = -\frac{\delta \tau m_{k,i}}{\sqrt{1 + \tau^2 m_{k,i}^2}}$, and

$$\theta_{k+1} = \theta_k - \delta \cdot \frac{\tau m_k}{\sqrt{1 + \tau^2 m_k^2}}, \tag{19}$$

applied coordinate-wise.

## A.2. Verification of Strong Convexity

**tanh regularizer.** The second derivative is

$$\frac{\partial^2 V}{\partial u^2}(u) = \frac{d}{du}\operatorname{arctanh}(u) = \frac{1}{1 - u^2}.$$

For $u \in (-1, 1)$, $\frac{1}{1-u^2} \geq 1$, so $V$ is 1-strongly convex on $(-1, 1)^d$.

**Algebraic regularizer.** The second derivative is

$$\frac{\partial^2 V}{\partial u^2}(u) = \frac{d}{du}\left[\frac{u}{\sqrt{1 - u^2}}\right] = \frac{1}{(1 - u^2)^{3/2}}.$$

For $u \in (-1, 1)$, $(1 - u^2)^{3/2} \leq 1$, so $\frac{\partial^2 V}{\partial u^2}(u) \geq 1$, and $V$ is 1-strongly convex on $(-1, 1)^d$.

In both cases Assumption 3.1 is satisfied, and $W_k(d) = \frac{\delta}{\tau}V(d/\delta)$ is $\frac{1}{\delta\tau}$-strongly convex on $(-\delta, \delta)^d$.

## A.3. Fenchel Conjugate

**tanh regularizer.** From the first-order condition $y = \operatorname{arctanh}(d)$, we have $d = \tanh(y)$. Using $1 \pm \tanh(y) = 2e^{\pm y}/(e^y + e^{-y})$:

$$V^*(y) = y\tanh(y) - \tfrac{1}{2}(1 + \tanh y)\ln(1 + \tanh y) - \tfrac{1}{2}(1 - \tanh y)\ln(1 - \tanh y)$$
$$= \ln(e^y + e^{-y}) - \ln 2 = \ln(\cosh(y)). \tag{20}$$

For the full regularizer, $V^*(y) = \sum_{i=1}^{d}\ln[\cosh(y_i)]$, and

$$W_k^*(y) = \frac{\delta}{\tau}V^*(\tau y) = \frac{\delta}{\tau}\sum_{i=1}^{d}\ln[\cosh(\tau y_i)]. \tag{21}$$

**Algebraic regularizer.** From the first-order condition $y = u/\sqrt{1 - u^2}$, we get $u = y/\sqrt{1 + y^2}$. Substituting into $V^*(y) = \sup_{u \in (-1,1)}\{yu - 1 + \sqrt{1 - u^2}\}$:

$$V^*(y) = \frac{y^2}{\sqrt{1 + y^2}} - 1 + \frac{1}{\sqrt{1 + y^2}} = \frac{y^2 + 1}{\sqrt{1 + y^2}} - 1 = \sqrt{1 + y^2} - 1. \tag{22}$$

For the full regularizer, $V^*(y) = \sum_{i=1}^{d}(\sqrt{1 + y_i^2} - 1)$, and

$$W_k^*(y) = \frac{\delta}{\tau}V^*(\tau y) = \frac{\delta}{\tau}\sum_{i=1}^{d}\left(\sqrt{1 + \tau^2 y_i^2} - 1\right). \tag{23}$$

# B. Missing Theoretical Details

In this section, we provide a detailed convergence analysis for the generalized momentum method (9). We first establish two key lemmas: one bounding the one-step progress (Lemma B.1), and another controlling the momentum error accumulation (Lemma B.2). Combining these results yields our main convergence theorem (Theorem 3.4).

## B.1. One-Step Descent Lemma

**Lemma B.1** (One-step progress)**.** *Suppose Assumptions 3.1 and 3.2 hold. Let $\|\cdot\|$ be an arbitrary norm with dual norm $\|\cdot\|_*$, and suppose $V$ is 1-strongly convex with respect to $\|\cdot\|$. Consider the update*

$$\theta_{k+1} = \theta_k + \arg\min_d\left\{\langle m_k, d\rangle + \frac{\delta}{\tau}V\left(\frac{d}{\delta}\right)\right\}. \tag{24}$$

*Let $\Delta\theta_k := \theta_{k+1} - \theta_k$, and $\xi_k := m_k - \nabla f(\theta_k)$ denote the momentum error. Then*

$$f(\theta_{k+1}) \leq f(\theta_k) - \frac{\delta}{\tau}V^*(-\tau\nabla f(\theta_k)) - \left(\frac{1}{2\delta\tau} - \frac{L}{2}\right)\|\Delta\theta_k\|^2 + \frac{\delta\tau}{2}\|\xi_k\|_*^2. \tag{25}$$

*Proof.* By $L$-smoothness of $f$,

$$f(\theta_{k+1}) \leq f(\theta_k) + \langle \nabla f(\theta_k), \Delta\theta_k \rangle + \frac{L}{2}\|\Delta\theta_k\|^2. \tag{26}$$

Define

$$W_k(d) := \frac{\delta}{\tau} V\left(\frac{d}{\delta}\right).$$

From the optimality condition for $\Delta\theta_k$, we have

$$\Delta\theta_k = \nabla W_k^*(-m_k),$$

which by Fenchel duality gives

$$W_k^*(-m_k) = \langle -m_k, \Delta\theta_k \rangle - W_k(\Delta\theta_k). \tag{27}$$

Rearranging,

$$\langle m_k, \Delta\theta_k \rangle = -\big(W_k(\Delta\theta_k) + W_k^*(-m_k)\big). \tag{28}$$

Since $\nabla f(\theta_k) = m_k - \xi_k$,

$$\langle \nabla f(\theta_k), \Delta\theta_k \rangle = \langle m_k, \Delta\theta_k \rangle - \langle \xi_k, \Delta\theta_k \rangle. \tag{29}$$

Substituting (28),

$$\langle \nabla f(\theta_k), \Delta\theta_k \rangle = -\big(W_k(\Delta\theta_k) + W_k^*(-m_k)\big) - \langle \xi_k, \Delta\theta_k \rangle. \tag{30}$$

Since $V$ is 1-strongly convex, $V^*$ is 1-smooth, hence

$$W_k^*(y) = \frac{\delta}{\tau} V^*(\tau y)$$

is $(\delta\tau)$-smooth. Applying smoothness with $u = -m_k$ and $v = -\nabla f(\theta_k)$,

$$W_k^*(-\nabla f(\theta_k)) \leq W_k^*(-m_k) + \langle \nabla W_k^*(-m_k), m_k - \nabla f(\theta_k) \rangle + \frac{\delta\tau}{2}\|m_k - \nabla f(\theta_k)\|_*^2. \tag{31}$$

Since $\nabla W_k^*(-m_k) = \Delta\theta_k$ and $m_k - \nabla f(\theta_k) = \xi_k$,

$$W_k^*(-\nabla f(\theta_k)) \leq W_k^*(-m_k) + \langle \Delta\theta_k, \xi_k \rangle + \frac{\delta\tau}{2}\|\xi_k\|_*^2. \tag{32}$$

Rearranging,

$$-W_k^*(-m_k) \leq -W_k^*(-\nabla f(\theta_k)) + \langle \Delta\theta_k, \xi_k \rangle + \frac{\delta\tau}{2}\|\xi_k\|_*^2. \tag{33}$$

Substituting (33) into (30),

$$\langle \nabla f(\theta_k), \Delta\theta_k \rangle \leq -W_k(\Delta\theta_k) - W_k^*(-\nabla f(\theta_k)) + \langle \Delta\theta_k, \xi_k \rangle + \frac{\delta\tau}{2}\|\xi_k\|_*^2 - \langle \xi_k, \Delta\theta_k \rangle \tag{34}$$

$$= -W_k(\Delta\theta_k) - W_k^*(-\nabla f(\theta_k)) + \frac{\delta\tau}{2}\|\xi_k\|_*^2. \tag{35}$$

Since $V$ is 1-strongly convex, $W_k$ is $\frac{1}{\delta\tau}$-strongly convex, hence

$$W_k(\Delta\theta_k) \geq \frac{1}{2\delta\tau}\|\Delta\theta_k\|^2.$$

Thus,

$$\langle \nabla f(\theta_k), \Delta\theta_k \rangle \leq -\frac{1}{2\delta\tau}\|\Delta\theta_k\|^2 - W_k^*(-\nabla f(\theta_k)) + \frac{\delta\tau}{2}\|\xi_k\|_*^2. \tag{36}$$

Finally, substituting this into (26) and recalling

$$W_k^*(-\nabla f(\theta_k)) = \frac{\delta}{\tau} V^*(-\tau\nabla f(\theta_k)),$$

we obtain the desired result. $\qquad\square$

## B.2. Momentum Error Bound

**Lemma B.2** (Momentum error recursion). *Suppose Assumptions 3.2 and 3.3 hold. Consider the momentum update*

$$m_k = \beta m_{k-1} + (1 - \beta) g_k, \tag{37}$$

*where $g_k$ is the stochastic gradient at iteration $k$. Let $\xi_k := m_k - \nabla f(\theta_k)$ denote the momentum error. Then*

$$\mathbb{E}[\|\xi_k\|_2^2] \leq \left(\frac{1 + \beta^2}{2}\right)^k \|\xi_0\|_2^2 + \frac{4\beta^2 L^2}{1 - \beta} \sum_{j=0}^{k-1} \left(1 - \frac{1 - \beta^2}{2}\right)^{k-1-j} \mathbb{E}[\|\Delta\theta_j\|_2^2] + 2(1 - \beta)\sigma^2. \tag{38}$$

*Moreover, for any $K \geq 1$, the following bounds hold:*

$$\sum_{k=0}^{K-1} \mathbb{E}[\|\xi_k\|_2^2] \leq \frac{2\|\xi_0\|_2^2}{1 - \beta^2} + \frac{4\beta^2 L^2}{(1 - \beta)^2} \sum_{k=0}^{K-1} \mathbb{E}[\|\Delta\theta_k\|_2^2] + 2K(1 - \beta)\sigma^2,$$

*Proof.* By definition,

$$\begin{aligned}
\xi_{k+1} &= m_{k+1} - \nabla f(\theta_{k+1}) \\
&= \beta m_k + (1 - \beta) g_{k+1} - \nabla f(\theta_{k+1}) \\
&= \beta \xi_k + \beta (\nabla f(\theta_k) - \nabla f(\theta_{k+1})) + (1 - \beta)(g_{k+1} - \nabla f(\theta_{k+1})).
\end{aligned} \tag{39}$$

Taking the squared norm,

$$\|\xi_{k+1}\|_2^2 = \|\beta\xi_k + \beta(\nabla f(\theta_k) - \nabla f(\theta_{k+1})) + (1 - \beta)(g_{k+1} - \nabla f(\theta_{k+1}))\|_2^2. \tag{40}$$

We expand this as

$$\begin{aligned}
\|\xi_{k+1}\|_2^2 = {} & \beta^2\|\xi_k + (\nabla f(\theta_k) - \nabla f(\theta_{k+1}))\|_2^2 + (1 - \beta)^2\|g_{k+1} - \nabla f(\theta_{k+1})\|_2^2 \\
& + 2\beta(1 - \beta)\langle\xi_k + (\nabla f(\theta_k) - \nabla f(\theta_{k+1})), g_{k+1} - \nabla f(\theta_{k+1})\rangle.
\end{aligned} \tag{41}$$

Since $\theta_k, \theta_{k+1}, m_k, g_k$ are all $\mathcal{F}_{k+1}$-measurable, taking conditional expectation $\mathbb{E}[\cdot \mid \mathcal{F}_{k+1}]$ and using $\mathbb{E}[g_{k+1} - \nabla f(\theta_{k+1}) \mid \mathcal{F}_{k+1}] = 0$ from Assumption 3.3, the cross-term vanishes:

$$\mathbb{E}[\|\xi_{k+1}\|_2^2 \mid \mathcal{F}_{k+1}] = \beta^2\|\xi_k + (\nabla f(\theta_k) - \nabla f(\theta_{k+1}))\|_2^2 + (1 - \beta)^2 \mathbb{E}[\|g_{k+1} - \nabla f(\theta_{k+1})\|_2^2 \mid \mathcal{F}_{k+1}]. \tag{42}$$

By Assumption 3.3, $\mathbb{E}[\|g_{k+1} - \nabla f(\theta_{k+1})\|_2^2 \mid \mathcal{F}_{k+1}] \leq \sigma^2$. For the first term, we expand:

$$\|\xi_k + (\nabla f(\theta_k) - \nabla f(\theta_{k+1}))\|_2^2 = \|\xi_k\|_2^2 + \|\nabla f(\theta_k) - \nabla f(\theta_{k+1})\|_2^2 + 2\langle\xi_k, \nabla f(\theta_k) - \nabla f(\theta_{k+1})\rangle.$$

By $L$-smoothness (Assumption 3.2),

$$\|\nabla f(\theta_k) - \nabla f(\theta_{k+1})\|_2^2 \leq L^2\|x_k - x_{k+1}\|_2^2 = L^2\|\Delta\theta_k\|_2^2. \tag{43}$$

For the cross-term, we use Young's inequality with parameter $a > 0$:

$$2\langle\xi_k, \nabla f(\theta_k) - \nabla f(\theta_{k+1})\rangle \leq a\|\xi_k\|_2^2 + \frac{1}{a}\|\nabla f(\theta_k) - \nabla f(\theta_{k+1})\|_2^2 \leq a\|\xi_k\|_2^2 + \frac{L^2}{a}\|\Delta\theta_k\|_2^2. \tag{44}$$

Combining these bounds,

$$\|\xi_k + (\nabla f(\theta_k) - \nabla f(\theta_{k+1}))\|_2^2 \leq (1 + a)\|\xi_k\|_2^2 + \left(1 + \frac{1}{a}\right) L^2\|\Delta\theta_k\|_2^2. \tag{45}$$

Taking full expectation,

$$\mathbb{E}[\|\xi_{k+1}\|_2^2] \le \beta^2(1+a)\mathbb{E}[\|\xi_k\|_2^2] + \beta^2\left(1+\frac{1}{a}\right)L^2\mathbb{E}[\|\Delta\theta_k\|_2^2] + (1-\beta)^2\sigma^2. \tag{46}$$

We choose $a = \frac{1-\beta^2}{2\beta^2}$. Then

$$\beta^2(1+a) = \beta^2\left(1+\frac{1-\beta^2}{2\beta^2}\right) = \beta^2 + \frac{1-\beta^2}{2} = \frac{2\beta^2+1-\beta^2}{2} = \frac{1+\beta^2}{2} = 1 - \frac{1-\beta^2}{2}. \tag{47}$$

For the second coefficient,

$$\beta^2\left(1+\frac{1}{a}\right)L^2 = \beta^2L^2 + \beta^2L^2 \cdot \frac{2\beta^2}{1-\beta^2} = \beta^2L^2\left(1+\frac{2\beta^2}{1-\beta^2}\right) = \beta^2L^2 \cdot \frac{1+\beta^2}{1-\beta^2}. \tag{48}$$

Thus,

$$\mathbb{E}[\|\xi_{k+1}\|_2^2] \le \left(1-\frac{1-\beta^2}{2}\right)\mathbb{E}[\|\xi_k\|_2^2] + \frac{\beta^2(1+\beta^2)L^2}{1-\beta^2}\mathbb{E}[\|\Delta\theta_k\|_2^2] + (1-\beta)^2\sigma^2. \tag{49}$$

Unrolling this recursion for $k \ge 0$,

$$\mathbb{E}[\|\xi_k\|_2^2] \le \left(1-\frac{1-\beta^2}{2}\right)^k\|\xi_0\|_2^2 + \frac{\beta^2(1+\beta^2)L^2}{1-\beta^2}\sum_{j=0}^{k-1}\left(1-\frac{1-\beta^2}{2}\right)^{k-1-j}\mathbb{E}[\|\Delta\theta_j\|_2^2]$$

$$+ (1-\beta)^2\sigma^2\sum_{j=0}^{k-1}\left(1-\frac{1-\beta^2}{2}\right)^{k-1-j}. \tag{50}$$

Since $\sum_{j=0}^{k-1}\left(1-\frac{1-\beta^2}{2}\right)^j \le \frac{2}{1-\beta^2}$, we obtain

$$\mathbb{E}[\|\xi_k\|_2^2] \le \left(1-\frac{1-\beta^2}{2}\right)^k\|\xi_0\|_2^2 + \frac{\beta^2(1+\beta^2)L^2}{1-\beta^2}\sum_{j=0}^{k-1}\left(1-\frac{1-\beta^2}{2}\right)^{k-1-j}\mathbb{E}[\|\Delta\theta_j\|_2^2] + \frac{2(1-\beta)^2\sigma^2}{1-\beta^2}.$$

Simplifying the coefficients, we note that $1-\beta^2 = (1-\beta)(1+\beta)$ and $1+\beta \le 2$, $1+\beta^2 \le 2$ for $\beta \in (0,1)$. Thus,

$$\frac{2\beta^2(1+\beta^2)}{1-\beta^2} \le \frac{2\beta^2 \cdot 2}{(1-\beta)(1+\beta)} \le \frac{4\beta^2}{(1-\beta) \cdot 1} = \frac{4\beta^2}{1-\beta}, \tag{51}$$

and

$$\frac{2(1-\beta)^2}{1-\beta^2} = \frac{2(1-\beta)^2}{(1-\beta)(1+\beta)} = \frac{2(1-\beta)}{1+\beta} \le \frac{2(1-\beta)}{1} = 2(1-\beta). \tag{52}$$

Therefore,

$$\mathbb{E}[\|\xi_k\|_2^2] \le \left(\frac{1+\beta^2}{2}\right)^k\|\xi_0\|_2^2 + \frac{4\beta^2L^2}{1-\beta}\sum_{j=0}^{k-1}\left(1-\frac{1-\beta^2}{2}\right)^{k-1-j}\mathbb{E}[\|\Delta\theta_j\|_2^2] + 2(1-\beta)\sigma^2. \tag{53}$$

We now proof the second inequality of Lemma B.2. Summing over $k = 0$ to $K-1$,

$$\sum_{k=0}^{K-1}\mathbb{E}[\|\xi_k\|_2^2] \le \sum_{k=0}^{K-1}\left(\frac{1+\beta^2}{2}\right)^k\|\xi_0\|_2^2 + \frac{4\beta^2L^2}{1-\beta}\sum_{k=0}^{K-1}\sum_{j=0}^{k-1}\left(1-\frac{1-\beta^2}{2}\right)^{k-1-j}\mathbb{E}[\|\Delta\theta_j\|_2^2] + 2K(1-\beta)\sigma^2. \tag{54}$$

We now deal with double sum. Let $\rho := 1 - (1 - \beta^2)/2 = (1 + \beta^2)/2 \in (0, 1)$. Then, using nonnegativity and changing the order of summation:

$$\sum_{k=0}^{K-1} \sum_{j=0}^{k-1} \rho^{k-1-j} \mathbb{E}\big[\|\Delta\theta_j\|_2^2\big]$$

$$= \sum_{j=0}^{K-2} \mathbb{E}\big[\|\Delta\theta_j\|_2^2\big] \sum_{k=j+1}^{K-1} \rho^{k-1-j} = \sum_{j=0}^{K-2} \mathbb{E}\big[\|\Delta\theta_j\|_2^2\big] \sum_{t=0}^{K-2-j} \rho^t$$

$$\leq \sum_{j=0}^{K-2} \mathbb{E}\big[\|\Delta\theta_j\|_2^2\big] \cdot \frac{1}{1-\rho} \leq \frac{1}{1-\rho} \sum_{j=0}^{K-1} \mathbb{E}\big[\|\Delta\theta_j\|_2^2\big].$$

Since $1 - \rho = \frac{1-\beta^2}{2}$, we obtain

$$\sum_{k=0}^{K-1} \sum_{j=0}^{k-1} \left(1 - \frac{1-\beta^2}{2}\right)^{k-1-j} \mathbb{E}\big[\|\Delta\theta_j\|_2^2\big] \leq \frac{2}{1-\beta^2} \sum_{j=0}^{K-1} \mathbb{E}\big[\|\Delta\theta_j\|_2^2\big] \leq \frac{2}{1-\beta} \sum_{j=0}^{K-1} \mathbb{E}\big[\|\Delta\theta_j\|_2^2\big].$$

Thus,

$$\sum_{k=0}^{K-1} \mathbb{E}[\|\xi_k\|_2^2] \leq \frac{2\|\xi_0\|_2^2}{1-\beta^2} + \frac{4\beta^2 L^2}{(1-\beta)^2} \sum_{j=0}^{K-1} \mathbb{E}[\|\Delta\theta_j\|_2^2] + 2K(1-\beta)\sigma^2. \tag{55}$$

$\square$

## B.3. Proof of the Main Theorem

**Theorem B.3** (Convergence rate). *Suppose Assumptions 3.1, 3.2, and 3.3 hold. Consider the momentum method with updates*

$$m_k = \beta m_{k-1} + (1-\beta)g_k, \quad \theta_{k+1} = \theta_k + \arg\min_d \left\{ \langle m_k, d \rangle + \frac{\delta}{\tau} V\left(\frac{d}{\delta}\right) \right\}. \tag{56}$$

*Let $\beta \in (0, 1)$ and the stepsize parameters satisfy*

$$\delta\tau \leq \frac{1}{2L} \cdot \min\left\{1; \frac{1-\beta}{\beta}\right\}.$$

*Then for any $K \geq 1$ it holds that:*

$$\frac{1}{K} \sum_{k=0}^{K-1} \frac{1}{\tau^2} \mathbb{E}[V^*(-\tau\nabla f(\theta_k))] \leq \frac{f(\theta_0) - f^*}{\delta\tau K} + (1-\beta)\sigma^2 + \frac{\|\nabla f(\theta_0)\|_2^2}{K(1-\beta)}. \tag{57}$$

*Proof.* From Lemma B.1 with Euclidean norm (as stated in Assumption 3.1), we have

$$f(\theta_{k+1}) \leq f(\theta_k) - \frac{\delta}{\tau} V^*(-\tau\nabla f(\theta_k)) - \left(\frac{1}{2\delta\tau} - \frac{L}{2}\right)\|\Delta\theta_k\|_2^2 + \frac{\delta\tau}{2}\|\xi_k\|_2^2. \tag{58}$$

Rearranging the terms,

$$\frac{\delta}{\tau} V^*(-\tau\nabla f(\theta_k)) \leq f(\theta_k) - f(\theta_{k+1}) - \left(\frac{1}{2\delta\tau} - \frac{L}{2}\right)\|\Delta\theta_k\|_2^2 + \frac{\delta\tau}{2}\|\xi_k\|_2^2. \tag{59}$$

Taking expectation and summing from $k = 0$ to $K - 1$, and using $f(\theta_K) \geq f^*$:

$$\sum_{k=0}^{K-1} \frac{\delta}{\tau} \mathbb{E}[V^*(-\tau\nabla f(\theta_k))] \leq f(\theta_0) - f^* - \sum_{k=0}^{K-1} \left(\frac{1}{2\delta\tau} - \frac{L}{2}\right)\mathbb{E}[\|\Delta\theta_k\|_2^2] + \sum_{k=0}^{K-1} \frac{\delta\tau}{2}\mathbb{E}[\|\xi_k\|_2^2]. \tag{60}$$

From Lemma B.2, we use the bound for the sum of momentum errors:

$$\sum_{k=0}^{K-1} \mathbb{E}[\|\xi_k\|_2^2] \leq \frac{2\|\xi_0\|_2^2}{1-\beta^2} + \frac{4\beta^2 L^2}{(1-\beta)^2} \sum_{k=0}^{K-1} \mathbb{E}[\|\Delta\theta_k\|_2^2] + 2K(1-\beta)\sigma^2. \tag{61}$$

Substituting this into the main inequality and collecting terms with $\mathbb{E}[\|\Delta\theta_k\|_2^2]$:

$$\sum_{k=0}^{K-1} \frac{\delta}{\tau} \mathbb{E}[V^*(-\tau\nabla f(\theta_k))] \leq f(\theta_0) - f^* + \sum_{k=0}^{K-1} \left( \frac{2\delta\tau\beta^2 L^2}{(1-\beta)^2} - \left( \frac{1}{2\delta\tau} - \frac{L}{2} \right) \right) \mathbb{E}[\|\Delta\theta_k\|_2^2]$$
$$+ \delta\tau K(1-\beta)\sigma^2 + \frac{\delta\tau\|\xi_0\|_2^2}{1-\beta^2}. \tag{62}$$

Under the stepsize condition $\delta\tau \leq \frac{1}{2L} \cdot \min\{1; \frac{1-\beta}{\beta}\}$, the coefficient of $\mathbb{E}[\|\Delta\theta_k\|_2^2]$ is non-positive. Thus, the summation terms vanish, and using $1 - \beta^2 \geq 1 - \beta$:

$$\sum_{k=0}^{K-1} \frac{\delta}{\tau} \mathbb{E}[V^*(-\tau\nabla f(\theta_k))] \leq f(\theta_0) - f^* + \delta\tau K(1-\beta)\sigma^2 + \frac{\delta\tau\|\xi_0\|_2^2}{1-\beta}. \tag{63}$$

Dividing by $K$ and using $\xi_0 = \nabla f(\theta_0)$, we obtain the final rate:

$$\frac{1}{K} \sum_{k=0}^{K-1} \frac{1}{\tau^2} \mathbb{E}[V^*(-\tau\nabla f(\theta_k))] \leq \frac{f(\theta_0) - f^*}{\delta\tau K} + (1-\beta)\sigma^2 + \frac{\|\nabla f(\theta_0)\|_2^2}{K(1-\beta)}. \tag{64}$$

$\square$

## C. Extension to General Norms

In this appendix, we extend the convergence analysis to the case of general norms, where the Bregman function $V$ is strongly convex with respect to an arbitrary norm $\|\cdot\|$ rather than the Euclidean norm. This generalization allows for a broader class of update rules, including those based on $\ell_1$, $\ell_\infty$, or other non-Euclidean geometries.

### C.1. Assumptions in General Norms

We begin by restating the key assumptions in the general norm setting.

**Assumption C.1** (Strong convexity of $V$ in general norm). The function $V : \mathbb{R}^d \to \mathbb{R}$ is 1-strongly convex with respect to a norm $\|\cdot\|$, i.e., for all $u, v \in \mathbb{R}^d$,

$$V(v) \geq V(u) + \langle \nabla V(u), v - u \rangle + \frac{1}{2}\|v - u\|^2. \tag{65}$$

**Assumption C.2** (Smoothness in general norm). The objective function $f : \mathbb{R}^d \to \mathbb{R}$ is $L$-smooth with respect to the dual norm $\|\cdot\|_*$, i.e., for all $\theta, \theta' \in \mathbb{R}^d$,

$$f(\theta') \leq f(\theta) + \langle \nabla f(\theta), \theta' - \theta \rangle + \frac{L}{2}\|\theta' - \theta\|_*^2. \tag{66}$$

**Assumption C.3** (Stochastic gradients in general norm). Let $\{\mathcal{F}_k\}_{k \geq 0}$ be a filtration such that $\theta_k$ is $\mathcal{F}_k$-measurable. At each iteration $k$ we have access to a stochastic gradient $g_k$ satisfying

$$\mathbb{E}[g_k \mid \mathcal{F}_k] = \nabla f(\theta_k), \qquad \mathbb{E}[\|g_k - \nabla f(\theta_k)\|^2 \mid \mathcal{F}_k] \leq \sigma^2. \tag{67}$$

## C.2. Analysis With Momentum via Norm Equivalence

Since all norms on $\mathbb{R}^d$ are equivalent, for any norm $\|\cdot\|$ there exists a constant $\rho_d \geq 1$ depending on $\|\cdot\|$ and $d$ such that

$$\|x\|_2 \leq \rho_d \|x\| \quad \text{for all } x \in \mathbb{R}^d. \tag{68}$$

By duality, (68) implies $\|x\|_* \leq \rho_d \|x\|_2$ for all $x$.

**Theorem C.4** (Convergence with momentum in general norm). *Under Assumptions C.1, C.2, C.3, and the norm equivalence* (68), *consider the momentum method with updates*

$$m_k = \beta m_{k-1} + (1 - \beta) g_k,$$

$$\theta_{k+1} = \theta_k + \arg\min_{d \in \delta \mathcal{D}} \left\{ \langle m_k, d \rangle + \frac{\delta}{\tau} V\left(\frac{d}{\delta}\right) \right\}.$$

*Let $\beta \in (0, 1)$ and*

$$\delta\tau \leq \frac{1}{2L} \cdot \min\left\{ 1 \; ; \; \frac{1 - \beta}{\rho_d^2 \beta} \right\}.$$

*Then for any $K \geq 1$,*

$$\frac{1}{K} \sum_{k=0}^{K-1} \frac{1}{\tau^2} \mathbb{E}[V^*(-\tau \nabla f(\theta_k))] \leq \frac{f(\theta_0) - f^*}{\delta\tau K} + \rho_d^2 (1 - \beta)\sigma^2 + \frac{\rho_d^2 \|\nabla f(\theta_0)\|_2^2}{K(1 - \beta)}. \tag{69}$$

*Proof.* The proof follows the same steps as Theorem B.3, with two applications of (68).

From Lemma B.1 and $\|\xi_k\|_*^2 \leq \rho_d^2 \|\xi_k\|_2^2$,

$$\frac{\delta}{\tau} \mathbb{E}[V^*(-\tau \nabla f(\theta_k))] \leq f(\theta_k) - f(\theta_{k+1}) - \left(\frac{1}{2\delta\tau} - \frac{L}{2}\right) \|\Delta\theta_k\|^2 + \frac{\delta\tau\rho_d^2}{2} \|\xi_k\|_2^2.$$

Summing over $k = 0, \ldots, K - 1$ and applying Lemma B.2 with $\|\Delta\theta_k\|_2^2 \leq \rho_d^2 \|\Delta\theta_k\|^2$,

$$\sum_{k=0}^{K-1} \frac{\delta}{\tau} \mathbb{E}[V^*] \leq f(\theta_0) - f^* + \sum_{k=0}^{K-1} \left(\frac{2\delta\tau\rho_d^4 \beta^2 L^2}{(1 - \beta)^2} - \left(\frac{1}{2\delta\tau} - \frac{L}{2}\right)\right) \mathbb{E}[\|\Delta\theta_k\|^2]$$

$$+ \delta\tau\rho_d^2 K(1 - \beta)\sigma^2 + \frac{\delta\tau\rho_d^2 \|\nabla f(\theta_0)\|_2^2}{1 - \beta}.$$

Under the stated stepsize condition the coefficient of $\mathbb{E}[\|\Delta\theta_k\|^2]$ is non-positive, so those terms drop out. Dividing by $\delta\tau K/\tau$ yields (69). $\square$

Setting $\rho_d = 1$ (i.e., $\|\cdot\| = \|\cdot\|_2$) recovers Theorem B.3 exactly. For a general norm, the rate degrades by a factor of $\rho_d^2$, which can grow with the dimension $d$. To avoid this constant, one would need an analogue of Lemma B.2 that bounds the momentum error directly in $\|\cdot\|_2$ without passing through the primal norm $\|\cdot\|$. Such a bound requires structural properties of the norm that are not available in the fully general setting. We therefore turn to the momentum-free case in the next subsection, where the norm equivalence argument is not needed and $\rho_d$ does not appear in the final rate.

## C.3. Analysis Without Momentum

For this reason, we focus on the momentum-free case, where the update rule simplifies to

$$\theta_{k+1} = \theta_k + \arg\min_d \left\{ \langle g_k, d \rangle + \frac{\delta}{\tau} V\left(\frac{d}{\delta}\right) \right\}. \tag{70}$$

In this setting, we directly apply Assumption C.3 to bound the stochastic noise term $\mathbb{E}[\|g_k - \nabla f(\theta_k)\|_*^2] \leq \sigma^2$.

**Theorem C.5** (Convergence without momentum in general norm). *Under Assumptions 3.1, 3.2, and 3.3, let $\delta\tau \leq \frac{1}{L}$. Then for any $K \geq 1$,*

$$\frac{1}{K} \sum_{k=0}^{K-1} \frac{1}{\tau^2} \mathbb{E}[V^*(-\tau\nabla f(\theta_k))] \leq \frac{f(\theta_0) - f^*}{\delta\tau K} + \frac{\sigma^2}{2}. \tag{71}$$

*Proof.* Let us recall Lemma B.1 for arbitrary norms. Let $\Delta\theta_k := \theta_{k+1} - \theta_k$, and $\varepsilon_k := g_k - \nabla f(\theta_k)$. Then

$$f(\theta_{k+1}) \leq f(\theta_k) - \frac{\delta}{\tau} V^*(-\tau\nabla f(\theta_k)) - \left(\frac{1}{2\delta\tau} - \frac{L}{2}\right) \|\Delta\theta_k\|^2 + \frac{\delta\tau}{2} \|\varepsilon_k\|_*^2. \tag{72}$$

Summing (72) over $k = 0, \ldots, K-1$, using $\delta\tau \leq 1/L$ (which implies $\frac{1}{2\delta\tau} - \frac{L}{2} \geq 0$) and $\mathbb{E}[\|\varepsilon_k\|_*^2] \leq \sigma^2$:

$$\mathbb{E}[f(\theta_K)] \leq f(\theta_0) - \frac{\delta}{\tau} \sum_{k=0}^{K-1} \mathbb{E}[V^*(-\tau\nabla f(\theta_k))] + \frac{\delta\tau K \sigma^2}{2}. \tag{73}$$

Since $f(\theta_K) \geq f^*$, rearranging the terms gives

$$\frac{\delta}{\tau} \sum_{k=0}^{K-1} \mathbb{E}[V^*(-\tau\nabla f(\theta_k))] \leq f(\theta_0) - f^* + \frac{\delta\tau K \sigma^2}{2}. \tag{74}$$

Dividing both sides by $\delta\tau K$ and multiplying by $\frac{1}{\tau}$, we obtain the final rate:

$$\frac{1}{K} \sum_{k=0}^{K-1} \frac{1}{\tau^2} \mathbb{E}[V^*(-\tau\nabla f(\theta_k))] \leq \frac{f(\theta_0) - f^*}{\delta\tau K} + \frac{\sigma^2}{2}. \tag{75}$$

$\square$

### C.4. Discussion: Convergence to a $\sigma$-Neighborhood

Unlike the momentum-based analysis in the Euclidean case (Theorem 3.4), where the variance reduction effect allows convergence to an $\mathcal{O}(\sigma^2/\sqrt{K})$ neighborhood of the optimum, the general norm analysis without momentum only guarantees convergence to a $\sigma$-neighborhood. Specifically, the term $\frac{\sigma^2}{2}$ in Theorem C.5 does not vanish as $K \to \infty$, reflecting the fundamental limitation of stochastic gradient methods without variance reduction.

However, this limitation can be mitigated in practice through mini-batching: by averaging gradients over a batch of size $B$, the effective noise level is reduced to $\sigma/\sqrt{B}$, allowing the algorithm to approach the optimum more closely. This is a standard technique in stochastic optimization and is widely used in deep learning applications.

In summary, while the general norm framework sacrifices the variance reduction benefits of momentum, it provides a flexible foundation for analyzing a broader class of optimization methods, including those based on non-Euclidean geometries that may be better suited to specific problem structures.

## D. Softmax Unigram Model

We use a softmax unigram model as a minimal next-token prediction setting where the degree of class imbalance can be controlled explicitly. This experiment is motivated by Yadav et al. (2025), who show that sign-based methods can converge faster than normalized gradient descent under heavy-tailed class imbalance. We follow their setup, but introduce a power-law parameter $\beta$ controlling token frequencies:

$$p_k \propto \frac{1}{k^\beta}.$$

Varying $\beta$ interpolates between nearly uniform and strongly heavy-tailed token distributions.

We compare three instances of `SoftSignum` (Algorithm 2): an SGD-like clipped regime, obtained with $\alpha_{\text{sign}} = 0$ and constant temperature $\mathcal{T} \equiv 1$; a pure sign regime, obtained with $\alpha_{\text{sign}} = 1$; and the full `SoftSignum` schedule combining Algorithms 2 and 1. Further training details are given in Appendix E.6.

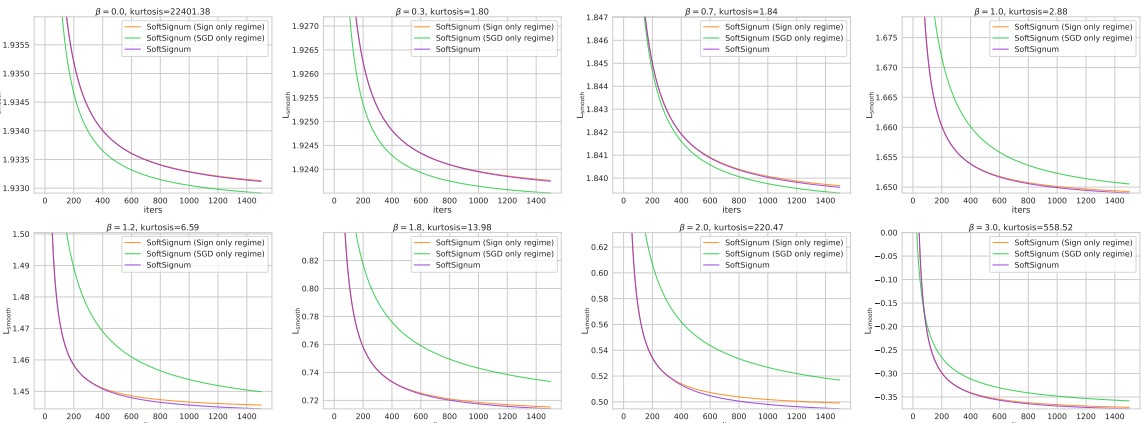

*Figure 5.* Unigram softmax optimization under power-law targets. The `SoftSignum` (SGD-only regime) curve uses $\alpha_{\text{sign}} = 0$ and constant temperature $\mathcal{T} \equiv 1$. The `SoftSignum` (Sign-only regime) curve uses $\alpha_{\text{sign}} = 1$. The `SoftSignum` curve uses the full temperature schedule from Algorithm 1.

Figure 5 shows that when the distribution is close to uniform, the SGD-like regime is sufficient and sign updates provide little benefit. As the distribution becomes more heavy-tailed, sign-based updates become increasingly useful. Across a broad intermediate range of $\beta$, the full `SoftSignum` schedule outperforms both fixed regimes, indicating that a smooth transition from sign-based to magnitude-sensitive updates is most beneficial under moderate heavy-tailed imbalance. For extremely heavy-tailed distributions, the gain over pure sign updates becomes smaller.

# E. Missing Experimental Details

## E.1. Next-character prediction Experimental Details

**Data processing.** For each sample, we extract text from both the *premise* and *hypothesis* fields of the NLI corpus (Laurer et al., 2022), including their original-language variants when available. Texts are filtered using a strict character-level criterion: a text is retained only if *all* its characters belong to a language-specific alphabet consisting of Latin letters, digits, whitespace, and punctuation. For Spanish and Italian, the alphabet additionally includes a small set of language-specific accented characters. After filtering, all retained texts from all languages are merged into a single unified corpus, without preserving language identifiers or enforcing language balance.

A restricted character vocabulary is constructed from the resulting corpus. Each text is segmented into sequences of length 50 characters using a sliding window with stride 12. For each input sequence, the corresponding target sequence is obtained by shifting the input by one character. The dataset is randomly split at the text level into training, validation, and test sets with proportions 80% / 10% / 10%, respectively.

**Training neural network.** All models described in the main part of the paper are trained for 50 epochs with batch size 64 using cross-entropy loss. Training relies solely on the optimizers specified in the main section of the paper, without further modifications. Performance is evaluated using character-level accuracy.

**Hyperparameter tuning** We perform hyperparameter tuning with the TPE sampler from the Optuna package (Akiba et al., 2019). We run 100 optimization trials with 20 startup trials. The objective function maximizes the accuracy score on the validation set, selecting the best epoch for each trial based on highest validation accuracy score. We tune learning rate, weight decay, momentum. To reduce run-time during tuning, we use a subset of 1000 randomly selected English, Spanish, and Italian texts from the NLI corpus (Laurer et al., 2022). Final evaluation with the best hyperparameters is performed on 5000 texts.

## E.2. GNN Training Experimental Details

**Model description.** Our GNN implementation features several enhancements: residual connections, dropout regularization, two-layer MLPs (instead of single linear layers) after each neighborhood aggregation step, and separate representations for ego- and neighbor-embeddings during aggregation (Platonov et al., 2023). We use fixed depth 3 and width 512.

**Training neural networks.** We use cross-entropy for as our loss function for classification and mean squared error for regression. We do not implement learning rate schedules, data augmentation techniques, or specialized neighbour sampling methods. Models are trained for 1000 epochs. Our experiments were conducted on a machine with NVIDIA A100 GPU with 80GB of VRAM.

**Optimizers implementation.** For `Muon` and `SoftMuon` we use these optimizers for all 2-d parameters, excluding input and output linear layers. For all other parameters (including 1-d, e.g. normalizations) we use `AdamW`.

**Hyperparameter tuning.** We conduct hyperparameter optimization using the TPE sampler with 50 iterations as implemented in the Optuna package (Akiba et al., 2019). For each model and dataset combination, we optimize a comprehensive set of hyperparameters including: regularization (dropout: 0.0-0.5, weight decay: 0.0-0.1), optimization parameters (learning rate: 0.0001-0.01 (for `Muon` based methods we independently tune learning rate for `AdamW` and `Muon`), momentum: 0.8-0.99). We use $\alpha_{\text{sign}} = 0.9$.

**Evaluation.** For each model and dataset combination, we evaluate the tuned hyperparameters across 8 different random seeds to ensure robustness of our results. As metric we use ROC-AUC for binary classification tasks and $R^2$ for regression. It is important to note that the data split remains consistent throughout both the tuning and evaluation phases to prevent tuning on any test node.

### E.3. LLM pretraining

For this section we use open source code from (Zmushko et al., 2024) (`https://github.com/fzmushko/FRUGAL`) and straightforward `SoftSignum` and `SoftMuon` integration.

**Training details.** We use batch size 512, sequence length 256, cosine-annealing learning rate scheduling with 10% warmup, and final learning rate 10 times lower than peak one. Our experiments were conducted on a machine with $8 \times$ NVIDIA A100 GPU with 80GB of VRAM.

*Table 4.* LLaMa 130M pretraining on C4 dataset. Final validation perplexity, averaged over 3 runs.

| SoftMuon | Muon | SoftSignum | Signum | AdamW |
|---|---|---|---|---|
| $\mathbf{18.288 \pm 0.002}$ | $18.591 \pm 0.002$ | $\mathbf{18.519 \pm 0.002}$ | $18.688 \pm 0.002$ | $18.709 \pm 0.002$ |

*Table 5.* `AdamW` tuning grid, bold entries highlight best hyperparameters.

| Parameter | Grid |
|---|---|
| lr | $10^{-4}, 10^{-3.75}, 10^{-3.5}, 10^{-3.25}, \mathbf{10^{-3}}, 10^{-2.75}, 10^{-2.5}, 10^{-2.25}, 10^{-2}$ |
| weight decay | $0.0, \mathbf{0.1}, 0.5$ |
| $\beta_1$ | $0.87, \mathbf{0.9}, 0.95$ |
| $\beta_2$ | $0.99, \mathbf{0.999}, 0.9999$ |
| $\varepsilon$ | $10^{-8}$ |

*Table 6.* `Signum` tuning grid, bold entries highlight best hyperparameters.

| Parameter | Grid |
|---|---|
| lr | $10^{-4}, 10^{-3.75}, \mathbf{10^{-3.5}}, 10^{-3.25}, 10^{-3}, 10^{-2.75}, 10^{-2.5}, 10^{-2.25}, 10^{-2}$ |
| weight decay | $0.0, \mathbf{0.1}, 0.5$ |
| momentum | $0.87, \mathbf{0.9}, 0.95$ |
| dampening | $\mathbf{0.0}, 0.9, 0.95$ |

Note that `Muon` uses `AdamW` for 1d parameters, embeddings and LM-head. To ensure the same tuning budget we keep $\beta_2$ fixed since it has a safe default value according to Semenov et al. (2025).

**Hyperparameter tuning.** `AdamW`, `Signum`, `Muon` have the same tuning computational budget (243 configurations). For the tuned optimizers, the selected parameters were interior points of the grid. Our methods (`SoftSignum` and `SoftMuon`)

*Table 7.* `Muon` tuning grid, bold entries highlight best hyperparameters.

| Parameter | Grid |
|---|---|
| lr | $10^{-4}, 10^{-3.75}, 10^{-3.5}, 10^{-3.25}, \mathbf{10^{-3}}, 10^{-2.75}, 10^{-2.5}, 10^{-2.25}, 10^{-2}$ |
| weight decay | $0.0, \mathbf{0.1}, 0.5$ |
| momentum | $0.9, \mathbf{0.95}, 0.99$ |
| Muon-AdamW lr adjustment | RMS-scaling |
| AdamW $\beta_1$ | $0.87, \mathbf{0.9}, 0.95$ |
| AdamW $\beta_2$ | $0.999$ |
| AdamW $\varepsilon$ | $10^{-8}$ |

require zero additional tuning budget: we do not perform further tuning and take parameters from the base version and set $\alpha_{sign} = 0.9$.

### E.3.1. 360M SMOLLM2 ON FINEWEB-EDU

**Training details.** Batch size 512, sequence length 1024, 1 epoch with Chinchilla-optimal number of tokens for tuning (and 2 times Chinchilla-optimal number of tokens for evaluation), cosine-annealing lr scheduling with 10% warmup (i.e. setup from (Semenov et al., 2025; Wen et al., 2025))

*Table 8.* SmolLM2-360M pretraining on FineWeb-Edu. Final validation perplexity, averaged over 3 runs.

| | Muon | SoftMuon |
|---|---|---|
| val perplexity | $12.541 \pm 0.004$ | $\mathbf{12.387 \pm 0.004}$ |

We have properly tuned `Muon` using the following grid, which is close to grids used in (Semenov et al., 2025; Wen et al., 2025):

*Table 9.* `Muon` tuning grid (360M SmolLM2), bold entries highlight best hyperparameters.

| Parameter | Grid |
|---|---|
| lr | $0.001, \mathbf{0.002}, 0.003$ |
| weight decay | $0.0, \mathbf{0.1}, 0.5$ |
| momentum | $0.9, \mathbf{0.95}, 0.99$ |
| NS iterations | 5 |
| Muon-AdamW lr adjustment | RMS-scaling |
| AdamW $\beta_1$ | $0.9$ |
| AdamW $\beta_2$ | $0.999$ |
| AdamW $\varepsilon$ | $10^{-8}$ |

For `SoftMuon`, we reuse the tuned `Muon` hyperparameters and set $\alpha_{sign} = 0.9$, requiring no additional tuning.

### E.3.2. 720M LLAMA ON FINEWEB

**Training details.** We mostly follow training setup of Semenov et al. (2025). We use effective batch size $\approx$1M tokens (total batch size 1984, sequence length 512), 48k iterations, cosine-annealing lr scheduling with 2000 warmup steps, gradient clipping 0.1, dropout 0.0 and final lr 10 times lower than peak lr.

For `SoftMuon`, we use the same hyperparameters and set $\alpha_{sign} = 0.9$.

### E.4. Unbalanced CIFAR Experimental Details

**Data preprocessing.** The class imbalance setup is described in Section 5.2. For all optimizers the same preprocessing is used for fair comparison. We modify the images from the CIFAR-10 dataset (Krizhevsky et al., 2009) with normalization. No data augmentations are applied during training. The training set is split into training (80%) and validation (20%) subsets.

**Training neural network.** We employ a simple CNN consisting of three convolutional stages, ReLU activations, and batch normalization. We use cross-entropy as the loss function. The batch size is fixed at 128. The maximum number of epochs is set to 50. Model performance is evaluated using the F1 score on both validation and test sets. We do not apply learning rate schedules.

**Hyperparameter tuning.** Hyperparameter tuning is performed with the TPE sampler from the Optuna package (Akiba et al., 2019). For each optimizer, we run 100 optimization trials with 20 startup trials. The objective function maximizes the F1 score on the validation set, selecting the best epoch for each trial based on the highest validation F1 score. The hyperparameters tuned include learning rate, weight decay, momentum (for momentum-based optimizers), and optimizer-specific parameters such as $\alpha_{sign}$ for `SoftSignum`.

### E.5. Smoothness Analysis Experiment Details

Setup of the experiment follows the next-character prediction task described in detail in Appendix E.1.

**Hyperparameter tuning.** We use the tuned parameters from Section 4.1.

**Smoothness constant estimation.** We run training with tuned hyperparameters, with logging (15). As said in Section 5.3, the same mini-batch is reused for all smoothness evaluations.

### E.6. Softmax Unigram Model

**Experimental setup.** Following (Yadav et al., 2025), we consider the following minimization problem:

$$\min_{\theta \in \mathbb{R}^d} \text{KL}(p_\beta \,\|\, q_\theta), \tag{76}$$

where $d \in \mathbb{N}$, $q_\theta = \text{Softmax}(\theta)$ is our model, and $p_\beta$ is a target categorical distribution over $d$ classes with the probability mass function

$$\mathbb{P}(x = k) = \frac{k^{-\beta}}{\sum\limits_{j=1}^{d} j^{-\beta}}, \quad k = 1, \dots, d. \tag{77}$$

In our experiments, we train the model $q_\theta$ with $d = 1000$. We consider a set of values $\beta \in \{0, \ 0.3, \ 0.7, \ 1.0, \ 1.2, \ 1.8, \ 2.0, \ 3.0\}$ in order to cover both near-uniform regimes ($\beta \approx 0$) and strongly heavy-tailed regimes with large $\beta$. Model parameters are initialized as $\theta \sim \mathcal{N}(0, I_d)$. We use the cross-entropy loss function and full-batch optimization for 1500 iterations. For each optimizer considered in this experiment, we do not use weight decay or momentum in order to better represent the dependence on the switch hyperparameters.

**Hyperparameter tuning.** Hyperparameter tuning is performed using the TPE sampler from the Optuna package (Akiba et al., 2019). For each optimizer, we run 100 optimization trials, including 20 startup trials. The objective function minimizes

$$L_{\text{smooth}} = \frac{1}{T} \sum_{i=1}^{T} \log L_i, \tag{78}$$

where $L_i$ is the value of the loss function at the $i$-th iteration and $T = 1500$ is the total number of training iterations. This choice is made to improve the informativeness of the visualizations.

For `SoftSignum` (SGD only iterations) optimizer and `SoftSignum` optimizer, we tune only the learning rate. For `SoftSignum` (Sign only iterations), we reuse the learning rate selected for `SoftSignum` in order to better visualize the effect of switching behaviour.

## F. On the Role of Large Language Models in Deriving the Theory

The convergence analysis of the soft-sign update

$$\theta_{k+1} = \theta_k - \delta \, \tanh(\tau \, m_k) \tag{79}$$

did not yield to a direct argument. After several attempts that left us with partial bounds, we used a large language model (`Claude` Opus 4.5, Anthropic (Anthropic, 2025)) to enumerate possible analytic strategies. We treated each suggestion as a candidate route to verify rather than as a derivation.

**Off-target reductions.** A first group of suggestions discarded the non-linearity of $\tanh$ altogether. The model proposed two opposite simplifications: drop the saturation and analyse the linear update $\theta_{k+1} = \theta_k - \delta\tau m_k$ as plain momentum `SGD`, or replace $\tanh(\tau m_k)$ by $\mathrm{sign}(m_k)$ and analyse the resulting iteration as `Signum`. For each reduction the model produced quantitative conditions of validity, $|\tau m_k| \ll 1$ for the linear case and $\tau|m_k| \gg 1$ for the sign case, and corresponding tightness bounds on the approximation error. Both reductions are accurate inside their own regime, but neither describes the algorithm we study: within a single iteration of `SoftSignum`, different coordinates of $\tau m_k$ lie on different sides of the saturation threshold, and a global linearization or a global sign approximation discards the parameter-wise heterogeneity that motivates the method.

**Reductions to classical analyses we had already tried.** A second group of suggestions reproduced approaches we had already attempted on paper. All of them recast (79) as a perturbed gradient step and invoke the standard non-convex stochastic gradient analysis under $L$-smoothness and bounded variance (Nesterov, 2004). The three concrete proposals were:

(i) Sandwiching $\tanh$ between linear maps. The two-sided bound

$$|\tanh(\tau m)| \leq \min\{|\tau m|, 1\}, \qquad |\tanh(\tau m) - \tau m| \leq \tfrac{1}{3}|\tau m|^3 \quad (\,|\tau m| \leq 1\,)$$

controls the deviation of $\tanh(\tau m_k)$ from the linear step $\tau m_k$ and absorbs it as a controlled error.

(ii) Local Taylor expansion. The identity

$$\tanh(\tau m_k) = \tau m_k - \tfrac{1}{3}(\tau m_k)^3 + O\big((\tau m_k)^5\big)$$

represents the cubic remainder as a higher-order perturbation of an `SGD` step.

(iii) Decomposition into a sign step plus a residual. The identity

$$\tanh(\tau m_k) = \mathrm{sign}(m_k) - r_k(\tau, m_k), \qquad |r_k(\tau, m_k)| \leq 2\,e^{-2\tau|m_k|}$$

absorbs $r_k$ into the variance term of a sign-based descent lemma.

Each of (i)–(iii) admits a formal version, but none of them measures progress in the geometry of the update itself. The resulting rate inevitably carries a factor of $\tau$ or $1/\tau$, and the bound collapses to that of momentum `SGD` in the limit $\tau \to 0$ or to that of `Signum` in the limit $\tau \to \infty$. The intermediate regime, in which different coordinates of $\tau m_k$ lie on different sides of the saturation threshold, is invisible to such analyses.

**The proximal-step reformulation.** One suggestion was qualitatively different. The model proposed to view (79) not as a perturbed gradient step but as a proximal step

$$\theta_{k+1} = \theta_k + \delta \arg\min_{d \in \mathcal{D}} \left\{ \langle m_k, d \rangle + \tfrac{1}{\tau} V(d) \right\} \tag{80}$$

for a 1-strongly convex regularizer $V$. This is exactly the general update (9) of Section 3. The natural analytic tool becomes Fenchel duality. Under Assumption 3.1 the conjugate $V^*$ is 1-smooth, and the optimality condition for (80) yields

$$\theta_{k+1} - \theta_k = \delta \nabla V^*(-\tau\, m_k). \tag{81}$$

Progress is then measured by $V^*(-\tau \nabla f(\theta_k))$ in place of a fixed norm of the gradient. The one-step descent estimate of Lemma B.1 and the convergence bound of Theorem 3.4 follow from this geometry without ad-hoc constants. Routes (i)–(iii) above are recovered as the Euclidean special case $V(u) = \tfrac{1}{2}\|u\|_2^2$, in which $V^*(y) = \tfrac{1}{2}\|y\|_2^2$ and the criterion reduces to the standard squared gradient norm.

**An algebraic error in the model's derivation.** In its own attempt to push the reformulation through to a concrete $V$, the model treated $\tanh(\alpha x)$ as $\alpha \tanh(x)$ and propagated this false linearity through several pages of computation. The candidate $V$ it returned was therefore inconsistent with (79). A single follow-up prompt asking the model to recheck its derivation was sufficient: the model located the offending step, retracted the linearity assumption, and reported the candidate $V$ as invalid. The reformulation idea itself survived this revision unchanged.

**What we did ourselves.** The model's contribution stopped at the high-level reformulation. The technical content of the paper is the authors' own. We identified the two regularizers whose first-order optimality conditions reproduce the $\tanh$ and

the algebraic soft-sign updates (Appendix A),

$$V_{\text{tanh}}(u) = \tfrac{1}{2} \sum_{i=1}^{d} \big[(1+u_i)\ln(1+u_i) + (1-u_i)\ln(1-u_i)\big], \qquad V_{\text{alg}}(u) = \sum_{i=1}^{d}\big(1 - \sqrt{1-u_i^2}\big),$$

verified that both are 1-strongly convex on $(-1,1)^d$ (Assumption 3.1), and computed the corresponding Fenchel conjugates in closed form

$$V_{\text{tanh}}^{*}(y) = \sum_{i=1}^{d} \ln\cosh(y_i), \qquad V_{\text{alg}}^{*}(y) = \sum_{i=1}^{d}\big(\sqrt{1+y_i^2} - 1\big).$$

The matrix extension and its convergence guarantee (Section 3.4) rest on the spectral lift (11) and on the conjugacy identity of Lewis (1996) reproduced in (13), both of which we identified as the right tools and adapted to the present setting. The one-step descent estimate (Lemma B.1), the momentum-error recursion (Lemma B.2), the proof of Theorem 3.4, the iteration-complexity statement of Corollary 3.5, the heterogeneous-regime interpretation of $V^{*}$ as $\sum_i \ln\cosh(\cdot)$ and the corresponding mixed $\ell_1$-vs-$\ell_2$ progress measure, and the general-norm treatment of Appendix C are written by the authors. On the algorithmic side, the quantile-based temperature schedule (Algorithm 1), the choice of the folded Cauchy distribution for momentum magnitudes, the practical defaults, the design and implementation of every experiment, and the local-smoothness analysis of Section 5.3 are likewise ours.

**Summary.** The role of the LLM in this work was that of a fast generator of analytic strategies. From the set of candidates it produced, one (the proximal-step reformulation (80)) provided the structural insight that organizes the convergence analysis. The remaining suggestions either reduced (79) to one of its two limiting regimes, repeated classical SGD-style routes that we had already tried, or contained algebraic errors. Selecting the surviving idea, identifying the correct regularizers, computing their Fenchel conjugates, lifting the construction to the matrix case, and carrying out the convergence analysis were the authors' own work. We view the model here as a research aid that accelerates the search over analytic strategies, not as a contributor of mathematical content.

