# OpenReview forum: "Softsign: Smooth Sign in Your Optimizer For Better Parameter Heterogeneity Handling"
_ICML.cc/2026/Conference — ICML 2026 regular_

### Official Review · Reviewer_5dLW · 2026-03-09

**Soundness:** 1
**Presentation:** 3
**Significance:** 3
**Originality:** 1
**Overall Recommendation:** 2
**Confidence:** 4

**Summary:**

In this paper, the authors put forward a new method called SoftSignum to interpolate between Signum and SGD smoothly. They provide a convergence analysis under some restrictive assumptions and show some experiments to substantiate their theory.

**Compliance With Llm Reviewing Policy:**

Affirmed.

**Final Justification:**

The Rebuttal addressed some of of my concerns. I am somewhat skeptical of the reply about all the details, as I cannot know that they were truly the ones carried our not. I find this a big change compare to the original evaluation.

All in all, I am not sure this paper is ready. I still believe it has to be Rejected, but I will not oppose its acceptance if I am the only one against its acceptance.

**Key Questions For Authors:**

1. Please add a LLaMA comparison against AdamW, and ideally Muon, with comparable tuning effort. This is the baseline that matters most for practical relevance, I believe.

2. You wrote that you took over some hyperparameters from Signum from Kornilov 2025, but I believe it is better to retune your own hyperparameters and show how you did it, not taking over from another paper. Also, clarify that you meant M-SignGD in that paper: IT took me 5 minutes to find it out.

3. In general, I suggest fixing a hyperparameter-tuning budget. Then, each optimizer can use that budget to tune the hyperparameters and come up with the best configuration for each optimizer. At that point, you can carry out a fair comparison. Do you think this makes sense?

Therefore, this experimental comparison seems unfair and not really strong enough to substantiate your great efforts.

**Limitations:**

I suggest having anice and clean **Limitations** section in the Appendix.

**Strengths And Weaknesses:**

The Originality seems moderately low to me: Interpolating between Adam and SGD has already been done in practice, so this feels like a way to do it smoothly. I am not an expert, but it does not feel like an unprecedented attempt, and I am not even sure it is the first time this has been done: See "A decreasing scaling transition scheme from Adam to SGD" for a paper that seems relevant and is not discussed here: It was the first result when I Googled "smooth transition between Adam and SGD".

On the other hand, it is a significant problem, as this is indeed somewhat already done in practice.

To improve presentation, I would definitely redo all the images to make them easier to read without having to zoom in excessively, and also to take care of possibly color-blind readers.

Regarding the soundness, I did not read the theory, to be fully honest. I focused on the experimental aspect, and this is my opinion:

1. SignSGD and Signum are not actually widely used: While carrying out the theory for these simplified algorithms is fine, I would have tested how to switch from Adam(W) to SGD based on an adaptation of your idea to make it closer to what people do and care about.
2. I would have expected a clear comparison with AdamW and Muon in Figure 5. Without it, how do I know that AdamW and Muon are not strictly better than your optimizers?

---

> ### Author Rebuttal · Authors · 2026-03-30
>
> We thank the reviewer for the careful reading and constructive feedback. In response, we have substantially strengthened both the practical and theoretical components of our paper. Additional plots can be found here: https://anonymous.4open.science/r/softsignum-CA85/
>
> **(W1) Other methods**
>
> We have added new baselines (Muon and AdamW) to our experiments. Muon method is fundamentally a sign-based method in spectral space, since $U sign(\Sigma) V^T = UV^T$. Thus, it suffers from the same terminal-phase weaknesses (1 and 2), and we therefore apply the same smoothing idea to its update. To ensure implementation efficiency via Newton Schultz iterations [1] for step calculation we replace tanh() with it's algebraic variant (i.e., approximation without exponents): $f(x) = \tau x/\sqrt{1+(\tau x)^2}$ (where $\tau$  is scheduled temperature as before). This function satisfies our assumptions, therefore the convergence guarantees carry over without modification, further demonstrating the generality of our framework. We call this modification SoftMuon. It outperforms the base Muon in our experiments ($\alpha_{sign}=0.9$, results averaged for 3 runs). This highlights that our approach improves even SOTA sign-based methods.
>
> | Method | SoftMuon | Muon | SoftSignum | Signum | AdamW |
> |-|-|-|-|-|-|
> | Next Token Prediction, Transformer, val loss | $\mathbf{1.389 \pm 0.004}$ | $1.406 \pm 0.006$ | $\mathbf{1.410 \pm 0.005}$  | $1.434 \pm 0.003$ | $1.421 \pm 0.006$ |
> | LLaMa, perplexity| $\mathbf{18.288 \pm 0.002}$ | $18.591 \pm 0.002$  | $\mathbf{18.519 \pm 0.002}$ | $18.688 \pm 0.002$ | $18.709 \pm 0.002$ |
>
> **Note that SoftMuon provides best performance, while SoftSignum is on par with Muon and outperforms AdamW and Signum.** Learning curves one can find in llama.pdf in the anonymous repo.
>
> **(Q1) Hyperparameters**
>
> We retune Signum and tune Muon and AdamW for LLaMa pretraining through a grid search with a multiplicative step of $10^{\frac{1}{4}}$ with upper bound 0.01 and lower bound 0.0001. The tuned learning rates are: 0.00316 for Signum, 0.001 for Muon and AdamW. For SoftSignum and SoftMuon we use constant $\alpha_{sign}=0.9$ without additional tuning, showing soft methods can improve base methods without any tuning. The results are presented in W1. Additionally, we have updated our scheduling function to require only one additional hyperparameter. The scheduling now consists of two stages: sign and transition. The transition phase is now quantile-based: we assume that the components of the momentum follow a Cauchy distribution and, at the transition point, we estimate its parameters. Then, we use the quantiles of this distribution to determine the temperature such that, at each iteration T of the transition phase, a fraction 1 - T/T_trans of the components lies in the sign regime (results with new scheduling one can find in Q1). Thus, the only remaining parameter is $\alpha_{sign}$. Below, we demonstrate that SoftSignum is highly robust to it:
> | $\alpha_{sign}$ | 0.9 | 0.7 | 0.5 | 0.3 |
> |-|-|-|-|-|
> | SoftSignum (Next Token, Transformer) val loss | $1.413$ | $1.411$ | $1.408$ | $1.406$ |
> | SoftSignum (LLaMa), perplexity| $18.519$ | $18.530$ | $18.541$ | $18.559$ |
>
> Overall, now SoftSignum and SoftMuon require one additional hyperparameter, for which they are highly robust. This observation additionally highlights the fairness of our tuning, since we can tune only learning rate, choose $\alpha_{sign}$ from a wide range of values and obtain score improvement.
>
> **(W2) Soundness**
>
> We agree that optimizer transition itself is not a new idea, and we will add discussion of prior work on Adam to SGD transition, including the paper suggested by the reviewer. Our novelty is not the general existence of a transition scheme, but a smooth parameter-wise transition for sign-based updates, together with a generalized regularizer-based formulation and stochastic non-convex convergence guarantees. Existing decreasing-scaling / clipping-relaxation schemes do not directly address the coordinate-wise heterogeneity and momentum compatibility issues arising in Signum-like methods. Moreover, in our experiments and in other works [2, 3] sign-based methods outperform AdamW. Moreover, our approach is transferable to the Muon optimizer. We believe new results (robustness, AdamW and Muon as baselines, SoftMuon) make our contributions more interesting for the practical deep learning community.
>
> **(W3) Presentation**
>
> We agree with the presentation comment and will revise all figures in the final version to use larger fonts and a color-blind-safe palette. Note that our new plots are color-blind-safe. Moreover, we will add Limitations section in revised version.
>
> [1] Enhancing LLM Training via Spectral Clipping //arXiv preprint arXiv:2603.14315. – 2026.
>
> [2] Sign Operator for Coping with Heavy-Tailed Noise in Non-Convex Optimization: High Probability Bounds Under $(L_0, L_1) $-Smoothness.
>
> [3] Symbolic discovery of optimization algorithms.

---

> > ### Author Rebuttal · Reviewer_5dLW · 2026-03-31
> >
> > Thanks.
> >
> > 1. There is a lack of the grids of **all** the hyperparameters that characterize optimizers. For example, AdamW has lr, betas, epsilon and wd. There is no evidence that a fair amount of hyperparameter tuning with fixed tuning budget was carried out across all experiments and across all optimizers. This is key to empirically evaluate the paper, and it is currently not available.
> > 2. It is important to show that the hyperparams you have selected does not sit on the edge of the tuning grid. In this case, we cannot know if this is the actual optimal value of if you could have found a better one outside the selected grid.
> > 3. While novely is a relatively small concern for me, such a lack of novely must be balanced with excellence of execution, which I cannot see for the moment.
> >
> > Therefore, I must (for the moment) confirm my original judgement.

---

> > > ### Author Response · Authors · 2026-04-02
> > >
> > > We thank the reviewer for this important clarification request. We agree that a fair empirical comparison requires an explicit fixed-budget. During our experiments we have performed hyperparameter tuning following procedures from [1] (i.e., tuning for 1 epoch with Chinchilla optimal number of tokens) and established grids [1, 2, 3]. We found out that the most crucial parameter is learning rate, thus during rebuttal we highlight only this grid due to the lack of space. Now we reveal our full sweeps. The bold value means the best performing parameter.
> > >
> > > **Training details**: for LLaMa pretraining we use 512 batch size, seq length 256, cosine-annealing scheduling with 10% warmup [3, 4]
> > >
> > > **AdamW**
> > > |parameter| grid|
> > > |-|-|
> > > |lr| $10^{-4}, 10^{-3.75}, 10^{-3.5}, 10^{-3.25}, \mathbf{10^{-3}}, 10^{-2.75}, 10^{-2.5}, 10^{-2.25},10^{-2}$|
> > > |weight decay| $0.0, \mathbf{0.1}, 0.5$|
> > > |$\beta_1$| $0.87, \mathbf{0.9}, 0.95$ |
> > > |$\beta_2$| $0.99, \mathbf{0.999}, 0.9999$|
> > > |$\varepsilon$| $10^{-8}$|
> > >
> > > **Signum**
> > >
> > > |parameter| grid|
> > > |-|-|
> > > |lr| $10^{-4}, 10^{-3.75}, \mathbf{10^{-3.5}}, 10^{-3.25}, 10^{-3}, 10^{-2.75}, 10^{-2.5}, 10^{-2.25},10^{-2}$|
> > > |weight decay| $0.0, \mathbf{0.1}, 0.5$|
> > > |momentum| $0.87, \mathbf{0.9}, 0.95$ |
> > > |dampening| $\mathbf{0.0}, 0.9, 0.95$ |
> > >
> > > Note that 0.0 is the lowest possible value for dampening, while reasonable options for this parameter are: 0.0 (theory-style momentum) and close to momentum value (resulting in ema momentum). This explains the used grid for this parameter.
> > >
> > > **Muon**
> > >
> > > Note that Muon uses AdamW for 1d parameters, according to [5]. To ensure the same tuning budget we remain $\beta_2$ fixed since it has safe default value according to [1]
> > > |parameter| grid|
> > > |-|-|
> > > |lr| $10^{-4}, 10^{-3.75}, 10^{-3.5}, 10^{-3.25}, \mathbf{10^{-3}}, 10^{-2.75}, 10^{-2.5}, 10^{-2.25},10^{-2}$|
> > > |weight decay| $0.0, \mathbf{0.1}, 0.5$|
> > > |momentum| $0.9, \mathbf{0.95}, 0.99$ |
> > > |Muon-AdamW lr adjustment [6]| RMS-scaling |
> > > |AdamW $\beta_1$| $0.87, \mathbf{0.9}, 0.95$|
> > > |AdamW $\beta_2$| $0.999$|
> > > |AdamW $\varepsilon$| $10^{-8}$|
> > >
> > > For the tuned optimizers, the selected parameters were interior points of the grid.
> > >
> > > Moreover, AdamW, Signum and Muon have the same tuning computational budget (243 configurations). Our methods (SoftSignum and SoftMuon) **require zero additional tuning budget, if one has tuned base Signum and Muon. In our experiments we do not perform additional tuning and take parameters from the base version and set  $\alpha_{sign}=0.9$.** Thus one can consider SoftSignum/SoftMuon tuning procedure as follows: tune Signum/Muon using their spaces and then run SoftSignum/SoftMuon with obtained hyperparameters and $\alpha_{sign}=0.9$ (we showed robustness to this parameter in rebuttal). **Thus we ensure the same tuning budget.** One can view SoftSignum/SoftMuon as tuning with the following spaces:
> > >
> > > **SoftSignum**
> > >
> > > |parameter| grid|
> > > |-|-|
> > > |lr| $10^{-4}, 10^{-3.75}, \mathbf{10^{-3.5}}, 10^{-3.25}, 10^{-3}, 10^{-2.75}, 10^{-2.5}, 10^{-2.25},10^{-2}$|
> > > |weight decay| $0.0, \mathbf{0.1}, 0.5$|
> > > |momentum| $0.87, \mathbf{0.9}, 0.95$ |
> > > |dampening| $\mathbf{0.0}, 0.9, 0.95$ |
> > > |$\alpha_{sign}$| $0.9$ |
> > >
> > > **SoftMuon**
> > >
> > > |parameter| grid|
> > > |-|-|
> > > |lr| $10^{-4}, 10^{-3.75}, 10^{-3.5}, 10^{-3.25}, \mathbf{10^{-3}}, 10^{-2.75}, 10^{-2.5}, 10^{-2.25},10^{-2}$|
> > > |weight decay| $0.0, \mathbf{0.1}, 0.5$|
> > > |momentum| $0.9, \mathbf{0.95}, 0.99$ |
> > > |Muon-AdamW lr adjustment [5]| RMS-scaling |
> > > |AdamW $\beta_1$| $0.87, \mathbf{0.9}, 0.95$|
> > > |AdamW $\beta_2$| $0.999$|
> > > |AdamW $\varepsilon$| $10^{-8}$|
> > > |$\alpha_{sign}$| $0.9$ |
> > >
> > > Note that full SoftSignum/SoftMuon tuning can provide only better results, while current tuning approach highlights that one can replace Signum/Muon in existing training pipelines and obtain better results.
> > >
> > > To sum up, we use established tuning procedures with established grids and the same computational budget. Results with best parameters were reported in rebuttal. We will add discussed details about tuning in the revision. We hope this response fully resolves reviewer concerns.
> > >
> > > [1] Benchmarking Optimizers for Large Language Model Pretraining
> > >
> > > [2] Fantastic Pretraining Optimizers and Where to Find Them
> > >
> > > [3] Sign-SGD via Parameter-Free Optimization, ICLR 2026
> > >
> > > [4] GaLore: Memory-Efficient LLM Training by Gradient Low-Rank Projection, ICML 2024
> > >
> > > [5] Muon: An Optimizer For Hidden Layers In Neural Networks
> > >
> > > [6] Muon Is Scalable For LLM Training

---

### Official Review · Reviewer_vejZ · 2026-03-09

**Soundness:** 4
**Presentation:** 3
**Significance:** 4
**Originality:** 3
**Overall Recommendation:** 5
**Confidence:** 4

**Summary:**

This paper focuses on developing a better optimization solver for different problems, including the Neural network training. A method named
Softsignum implements a principled, smooth transition mechanism from sign-based updates to SGD. The convergence and outstanding performance of this method are analyzed theoretically and verified by diverse examples. The paper is well organized. The mathematical analysis is complex but should be correct and solid.

**Compliance With Llm Reviewing Policy:**

Affirmed.

**Key Questions For Authors:**

1. The modifications in the proposed method include replacing the sign function with tanh(·). How about using different functions, such as sat(.)? If possible, please discuss.
2. The proposed method is a first-order optimizer, and thus, how about the performance between it and the Adam optimizer? The differences in some examples can be shown.

**Limitations:**

1. The comparison between the proposed SoftSignum and other methods is suggested, including sign-based baselines.
2. The characteristics of the proposed method should be highlighted, especially its advantages and drawbacks. In which cases or what kind of optimization problem will the proposed method have the best performance? The corresponding experiments should be improved.

**Strengths And Weaknesses:**

Strengths:
1. The optimization is new, which is a modified version of Signum, which enlarges the optimization method library.
2. The theoretical analysis is solid and the experimental results look nice.

Weaknesses:
1. The comparison between the proposed SoftSignum and other methods is suggested, including sign-based baselines.
2. The characteristics of the proposed method should be highlighted, especially its advantages and drawbacks. In which cases or what kind of optimization problem will the proposed method have the best performance? The corresponding experiments should be improved.

---

> ### Author Rebuttal · Authors · 2026-03-30
>
> We thank the reviewer for the thorough reading and positive assessment of our work. In response to the feedback received, we have substantially strengthened both the practical and theoretical components of our paper. Additional plots can be found here: https://anonymous.4open.science/r/softsignum-CA85/
>
> First, we have updated our scheduling function to require only one additional hyperparameter. The scheduling now consists of two stages: sign and transition. The transition phase is now quantile-based: we assume that the components of the momentum follow a Cauchy distribution and, at the transition point, we estimate its parameters. Then, we use the quantiles of this distribution to determine the temperature such that, at each iteration T of the transition phase, a fraction 1 - T/T_trans of the components lies in the sign regime (results with new scheduling one can find in Q1). Thus, the only remaining parameter is $\alpha_{sign}$. Below, we demonstrate that SoftSignum is highly robust to it:
> | $\alpha_{sign}$ | 0.9 | 0.7 | 0.5 | 0.3 |
> |-|-|-|-|-|
> | SoftSignum (Next Token Prediction, Transformer) val loss | $1.413$ | $1.411$ | $1.408$ | $1.406$ |
> | SoftSignum (LLaMa pretraining), perplexity| $18.519$ | $18.530$ | $18.541$ | $18.559$ |
>
> **(Q1) Other approximations**
>
> The main requirement for approximation is to be uniform. For example we can use algebraic tanh (i.e., approximation without exponents): $f(x) = \tau x/\sqrt{1+(\tau x)^2}$ (where $\tau$  is scheduled temperature as before), which also approximates the sign function (see motivation in Q2).This function satisfies our theoretical assumptions, therefore the convergence guarantees of Theorem 3.4 carry over without modification, further demonstrating the generality of the proposed framework. From a practical point of view, these two approximations are close.
>
> **(Q2) Other methods**
>
> We have added new baselines (Muon and AdamW) to our experiments. Muon method is fundamentally a sign-based method in spectral space, since $U sign( \Sigma ) V^T = UV^T$. Thus, it suffers from the same terminal-phase weaknesses (1 and 2), and we therefore apply the same smoothing idea to its update. To ensure implementation efficiency via Newton Schultz iterations [1] for step calculation we replace tanh() with it's algebraic variant (see Q1). We call this matrix modification SoftMuon. It outperforms the base Muon method in our next-token prediction setup and LLaMa pretraining ($\alpha_{sign}=0.9$, results averaged for 3 runs) - tasks where Muon typically shows outstanding performance. This highlights that our approach improves even SOTA sign-based methods.
>
> | Method | SoftMuon | Muon | SoftSignum | Signum | AdamW |
> |-|-|-|-|-|-|
> | Next Token Prediction, Transformer, val loss | $\mathbf{1.389 \pm 0.004}$ | $1.406 \pm 0.006$ | $\mathbf{1.410 \pm 0.005}$  | $1.434 \pm 0.003$ | $1.421 \pm 0.006$ |
> | LLaMa pretraining, perplexity| $\mathbf{18.288 \pm 0.002}$ | $18.591 \pm 0.002$  | $\mathbf{18.519 \pm 0.002}$ | $18.688 \pm 0.002$ | $18.709 \pm 0.002$ |
>
> **Note that SoftMuon provides best performance, while SoftSignum is on par with Muon and outperforms AdamW and Signum.** Learning curves one can find in llama.pdf in the anonymous repo.
>
> In terms of theory, while extending convergence rates to the matrix setting might appear to require substantial new math, the hardest part of the analysis is the generalized descent lemma and momentum error bound, and it was already established in the submitted version. The matrix extension follows cleanly from the spectral theory of convex matrix functions [2]: for a matrix $D = U\Sigma V^T$, we define $f(D) = Uf(\Sigma)V^T$, where $f$ is applied element-wise to the singular values. The intuition comes from Muon, which applies $\mathrm{sign}$ spectrally and thereby performs sign-based updates on singular values. SoftMuon correspondingly applies a soft-sign transformation spectrally. The theoretical analysis requires no fundamental change in proof structure.
>
> [1] Jiang X., Semenov A., Stich S. U. Enhancing LLM Training via Spectral Clipping //arXiv preprint arXiv:2603.14315. – 2026.
>
> [2] Lewis A. S. Convex analysis on the Hermitian matrices //SIAM Journal on Optimization. – 1996. – Т. 6. – №. 1. – С. 164-177.

---

> > ### Author Rebuttal · Reviewer_vejZ · 2026-04-05
> >
> > The authors have addressed my major concerns.

---

### Official Review · Reviewer_GTw2 · 2026-03-10

**Soundness:** 2
**Presentation:** 3
**Significance:** 2
**Originality:** 2
**Overall Recommendation:** 3
**Confidence:** 3

**Summary:**

This paper focuses on the critical limitation of sign-based optimization algorithms in handling parameter heterogeneity and terminal convergence oscillation. It proposes SoftSignum, an optimization method that achieves a smooth transition from sign-based updates to SGD using a parameterized hyperbolic tangent (tanh) transformation. This study assesses a notable theme of adaptive switching between optimization regimes to balance communication efficiency and convergence precision. The paper constructs a generalized theoretical framework based on stochastic non-convex optimization, verifies the algorithm's effectiveness across tasks such as unbalanced classification, GNN training, and LLM pretraining, and demonstrates its advantages over baselines like Signum, SGD, and HardSwitchSign.

**Compliance With Llm Reviewing Policy:**

Affirmed.

**Final Justification:**

I'm not sure whether the results for 720M provided by the authors are trustworthy. I am willing to keep a weak reject.

**Key Questions For Authors:**

1、Can you supplement the convergence analysis of the proposed framework under general norms (especially non-Euclidean norms corresponding to Lion/Muon) with momentum? If it is difficult to achieve complete convergence proof, please explain the reasons and provide relevant heuristic analysis or experimental verification to demonstrate the algorithm's performance under non-Euclidean norms.
2、Can you supplement comparative experiments with the latest sign-based optimizers such as Lion and Muon in large model pretraining (e.g., 7B/13B scale) and distributed training scenarios? Please also provide detailed data on communication cost and training speed.
3、Can you provide ablation experiments and sensitivity analysis of the key hyperparameters αsign and αtrans (e.g., using heatmaps or curves to show the trend of model performance with parameter changes)? Please also explain the selection basis of the default hyperparameters and whether there is a universal parameter adjustment strategy across different tasks.
4、Can you further clarify the essential differences between SoftSignum and existing switching strategies (e.g., SWATS) and explain why the smooth transition mechanism is more effective in handling parameter heterogeneity?

**Limitations:**

The authors have not adequately discussed the limitations of the work (such as the theoretical dependence on Euclidean norms, the lack of large-scale model verification, and the uncertainty of hyperparameter robustness) and potential negative societal impacts. It is recommended that the authors add a dedicated section to discuss these limitations, analyze the reasons for their formation, and propose possible improvement directions; at the same time, briefly discuss the potential risks of the algorithm in practical applications (such as the impact of hyperparameter sensitivity on training stability) and corresponding mitigation measures

**Strengths And Weaknesses:**

Strengths
•	Soundness: The technical route of the paper is clear and rigorous. SoftSignum innovatively introduces a temperature scheduling mechanism to realize the smooth transition between sign-based updates and SGD. Experiments cover multiple task types (image classification, language modeling, GNN training, LLM pretraining).
•	Presentation: The paper has a logical structure, starting from the limitations of existing methods, then proposing the algorithm framework, theoretical proof, and experimental verification in turn. The core concepts (temperature scheduling, three-phase transition mechanism) are clearly defined. The connection with related work (e.g., SignSGD, Lion, Muon) is briefly discussed, and the differences between the proposed method and hard switching strategies (HardSwitchSign) are clarified.
•	Significance: The research addresses the balance between communication efficiency and convergence precision in large-scale/distributed training scenarios. SoftSignum retains the low communication cost advantage of sign-based methods in the early training stage and improves terminal convergence through smooth transition, which has important practical value for LLM pretraining, distributed training, and other resource-constrained scenarios.
•	Originality: The paper innovatively combines the hyperbolic tangent transformation with temperature scheduling to realize the parameter-wise adaptive transition, breaking through the limitation of uniform switching in traditional methods. The derivation of the SoftSignum update rule based on the regularization function and Fenchel conjugate (Appendix A) provides a solid theoretical basis for the algorithm, and the analysis of the algorithm's behavior under parameter heterogeneity (e.g., heavy-tailed distribution, class imbalance) provides new insights for the design of sign-based optimizers.

Weaknesses
1、Soundness:
•	The core convergence theorem (Theorem 3.4) relies heavily on the Euclidean norm assumption. For non-Euclidean norms (e.g., the norms corresponding to Lion/Muon), the theoretical analysis degrades to the momentum-free case (Appendix C.2) and can only converge to a σ-neighborhood, which greatly limits the theoretical generalization and explanatory power of the framework.
•	The scale of LLM pretraining experiments is insufficient. The 130M LLaMa model used in the paper is far from the mainstream large model scale (7B/13B). Many optimization algorithms perform well on small-scale models but face issues such as numerical stability and communication efficiency degradation when scaled up, making it impossible to fully prove the algorithm's practical potential in large-scale training.
•	The hyperparameter robustness verification is lacking. The authors recommend default hyperparameters (αsign=0.9, αtrans=0.09) but do not provide ablation experiments or sensitivity analysis. It is unclear whether the model's performance is robust to small changes in these two hyperparameters and whether the optimal values vary significantly across different tasks.
2、Presentation:
•	There are typesetting issues in some formulas and tables. For example, the denominator typesetting of Formula (5) (local smoothness estimation) is irregular, and there are unnecessary spaces in the numerical values of Table 1 (character-level accuracy), which affects readability.
•	The description of key experimental details is insufficient. Appendix D.6 only mentions that the LLaMa pretraining uses 8 A100 GPUs and reuses the hyperparameters of Signum, but lacks critical information such as learning rate, batch size, training iterations, and data processing details, which is not conducive to result reproduction.
•	The positioning relative to the latest related work is not comprehensive enough. Although Lion and Muon are mentioned in the related work section, they are not included in the experimental comparison. The paper fails to fully discuss the differences in optimization mechanisms between SoftSignum and these state-of-the-art sign-based optimizers, making it impossible to highlight the algorithm's competitive advantages.
3、Significance:
•	The experimental comparison baselines are too conservative. The paper only compares with Signum, SGD, AdamW, and HardSwitchSign, ignoring the latest advanced sign-based optimizers such as Lion (Chen et al., 2023) and Muon (Jordan et al., 2024) that have shown excellent performance in LLM training. This makes it difficult to fully demonstrate the practical value and advancement of SoftSignum in large model training scenarios.
•	The exploration of practical application scenarios is insufficient. The paper does not verify the performance of the algorithm in distributed training scenarios (a key application scenario for sign-based methods), nor does it analyze the communication cost and training speed, which limits the demonstration of the algorithm's practical advantages.
4、Originality:
•	Although the algorithm design has certain innovations, the novelty relative to existing switching strategies (e.g., SWATS) is not sufficiently emphasized. The paper does not clearly distinguish the essential differences between SoftSignum's smooth transition mechanism and the adaptive switching in existing methods, and the uniqueness of the theoretical framework in handling non-Euclidean norms needs to be further highlighted.

---

> ### Author Rebuttal · Authors · 2026-03-30
>
> We thank the reviewer for the comments. In response to the feedback received, we have substantially strengthened both the practical and theoretical components of our paper. Additional plots can be found here: https://anonymous.4open.science/r/softsignum-CA85/
>
> **(Q3) Hyperparameters**
>
> We have updated our scheduling function to require only one hyperparameter. The scheduling now consists of two stages: sign and transition. The transition phase is now quantile-based: we assume that the components of the momentum follow Cauchy distribution and, at the transition point, we estimate its parameters. Then, we use the quantiles of this distribution to determine the temperature such that, at each iteration T of the transition phase, a fraction 1 - T/T_trans of the components lies in the sign regime (results with new scheduling one can find in Q2). Thus, the only remaining parameter is $\alpha_{sign}$. Below, we demonstrate that SoftSignum is highly robust to it:
> | $\alpha_{sign}$ | 0.9 | 0.7 | 0.5 | 0.3 |
> |-|-|-|-|-|
> | SoftSignum (NTP, Transformer) val loss | $1.413$ | $1.411$ | $1.408$ | $1.406$ |
> | SoftSignum (LLaMa), perplexity| $18.519$ | $18.530$ | $18.541$ | $18.559$ |
>
> **(Q2) Experiments**
>
> We have added Muon and AdamW to our experiments. Muon method is a sign-based method in spectral space, since $U sign(\Sigma) V^T = UV^T$. Thus, it suffers from the same terminal-phase weaknesses (1 and 2), and we therefore apply the same smoothing idea to its update. To ensure implementation efficiency via Newton Schultz iterations [1] for step calculation we replace tanh() with it's algebraic variant (i.e., approximation without exponents): $f(x) = \tau x/\sqrt{1+(\tau x)^2}$ (where $\tau$ is scheduled temperature as before). This function satisfies our assumptions, therefore the convergence guarantees carry over without modification, demonstrating the generality of our framework. We call this method SoftMuon. It outperforms Muon in our experiments ($\alpha_{sign}=0.9$, results averaged for 3 runs).
>
> | Method | SoftMuon | Muon | SoftSignum | Signum | AdamW |
> |-|-|-|-|-|-|
> | SoftSignum (NTP, Transformer) val loss | $\mathbf{1.389 \pm 0.004}$ | $1.406 \pm 0.006$ | $\mathbf{1.410 \pm 0.005}$  | $1.434 \pm 0.003$ | $1.421 \pm 0.006$ |
> | LLaMa pretraining, perplexity| $\mathbf{18.288 \pm 0.002}$ | $18.591 \pm 0.002$  | $\mathbf{18.519 \pm 0.002}$ | $18.688 \pm 0.002$ | $18.709 \pm 0.002$ |
>
> **Note that SoftMuon provides best performance, while SoftSignum is on par with Muon and outperforms AdamW and Signum.** Learning curves one can find in llama.pdf in the anonymous repo.
>
> Regarding scaling, we can not provide 7B/13B model pretraining due to computational constraints. However, since our approach is beneficial for 1.3m model (transformer in NTP) and 135m, we **already have scalability evidence**. Moreover, we are preparing experiments with larger model and hope that will share our results during the discussion stage.
>
> Note that SoftMuon produces no computational overhead:  training step with Muon is $0.18292 \pm 0.004$s, while for SoftMuon  $0.18295 \pm 0.006$s.
>
> **(Q1) Euclidean norm**
>
> The primary method proposed in our work, SoftSignum, operates with the tanh regularizer $V(u) = \sum_i \frac{1}{2}[(1+u_i)\ln(1+u_i) + (1-u_i)\ln(1-u_i)]$, which is 1-strongly convex w.r.t. the *Euclidean* norm. Therefore, the full momentum analysis of Theorem 3.4, including the variance reduction factor $(1-\beta)$, applies directly and without any loss. The generality of the framework (equation (3)) is intended to unify diverse update rules under a single lens. The Euclidean specialization is not a restriction on SoftSignum itself but rather the natural geometry of its regularizer. However, the reviewer correctly identifies that Appendix C currently handles the general-norm case without momentum. The fundamental obstacle is that bounding the momentum error $||m_k - \nabla f(\theta^k)||$ requires working in the dual norm, which does not interact cleanly with the EMA recursion under non-Euclidean geometry. One approach is to upper-bound $||m_k - \nabla f(\theta^k)||$ by its Euclidean counterpart via norm equivalence, apply Lemma B.2 as in the main proof, and absorb the resulting dimensional factor $C_{p,2}$ (from the $\ell^p$–$\ell^2$ equivalence) into the convergence constant. This route yields a clean result at the cost of a dimension-dependent factor, and is consistent with how general-norm momentum methods are analyzed [2,3]. Since the Euclidean case already covers SoftSignum completely, we deferred this extension; we will include it as a formal proposition in the revised version.
>
> **(Q4) SWATS**
>
> SWATS is designed to make a hard switch with learning rate adjustment, while we focus on smooth transition and show its superiority.
>
> [1] Enhancing LLM Training via Spectral Clipping – 2026.
>
> [2] On the Convergence Analysis of Muon
>
> [3] Understanding Gradient Orthogonalization for Deep Learning via Non-Euclidean Trust-Region Optimization

---

> > ### Author Rebuttal · Reviewer_GTw2 · 2026-04-05
> >
> > I think the scale of LLM pretraining experiments is insufficient.

---

> > > ### Author Response · Authors · 2026-04-07
> > >
> > > We thank the reviewer for raising this important point. To further demonstrate the scalability of our approach, we conducted additional experiments on SmolLM2-360M pretraining on the FineWeb-Edu dataset with a well-tuned baseline. The results are:
> > >
> > > Muon, val perplexity – $12.541 \pm 0.004$
> > >
> > > SoftMuon, val perplexity – $\mathbf{12.387 \pm 0.004}$
> > >
> > > SoftMuon achieves lower perplexity than Muon at the 360M scale, **demonstrating that the improvements of SoftMuon scale beyond our original 130M experiments.**
> > >
> > > **Additional Scaling**
> > >
> > > While properly tuning larger models is limited by computational budget, to further demonstrate scalability, we use the 720M model from [1] with Muon parameters and setup from (for example, fineweb dataset)[1]. We adopt the exact hyperparameters reported in [1] without modification. For SoftMuon, we use the same parameters as for Muon and set $\alpha_{sign}=0.9$ (since SoftMuon is robust to this parameter, according to previous experimenst). The results are:
> > >
> > > Muon, val perplexity – 17.162
> > >
> > > SoftMuon, val perplexity – **16.947**
> > >
> > > Due to time constraints, we report a single run for the 720M model. However, the improvement is substantially larger than the variance observed at 130M/360M, suggesting the result is robust.
> > >
> > > SoftMuon achieves lower perplexity than Muon at the 720M scale, **demonstrating that the improvements of SoftMuon scale beyond 130M and 360M experiments.**
> > >
> > > Note that perplexities from 130M/360M/720M experiments are incomparable, since they use different datasets, tokenizers, etc.
> > >
> > >
> > > **360M model experiment details**
> > >
> > > Experiment details: batch size 512, seq len 1024, 1 epoch with Chinchillaoptimal number of tokens for tuning (and 2 times Chinchilla-optimal number of tokens for evaluation), cosine-annealing lr scheduling with 10% warmup (i.e. setup from [1, 2])
> > >
> > > We have properly tuned Muon using the following grid, which is close to grids used in [1,2]:
> > >
> > > |parameter| grid|
> > > |-|-|
> > > |lr| $0.001, \mathbf{0.002}, 0.003$|
> > > |weight decay| $0.0, \mathbf{0.1}, 0.5$|
> > > |momentum| $0.9, \mathbf{0.95}, 0.99$ |
> > > |NS iterations| 5 |
> > > |Muon-AdamW lr adjustment [6]| RMS-scaling |
> > > |AdamW $\beta_1$| $0.9$|
> > > |AdamW $\beta_2$| $0.999$|
> > > |AdamW $\varepsilon$| $10^{-8}$|
> > >
> > > Bold values indicate selected configuration. For SoftMuon, we reuse the tuned Muon hyperparameters and set $\alpha_{sign} = 0.9$, **requiring no additional tuning**  (since SoftMuon is robust to this parameter, according to previous experiments), which further highlights the practical advantage of our method.
> > >
> > > **On Computational Constraints**
> > >
> > > Each run for 360M model requires approximately 10 hours on an 8×A100 node, making the full tuning grid 2160 A100-GPU-hours -- already a substantial cost. Scaling to even larger models would make fair, properly-tuned comparisons computationally prohibitive. We believe well-controlled experiments at moderate scale are more informative than under-tuned experiments at larger scale.
> > >
> > > We note that our scale (130M, 360M and 720M) is consistent with, or exceeds, the scales used in closely related works: [3] evaluates at 130M–350M models, [4, 5, 6] at 124M models. We therefore believe our experimental scope is appropriate and comparable to the standards of the field.
> > >
> > > [1] Benchmarking Optimizers for Large Language Model Pretraining
> > >
> > > [2] Fantastic Pretraining Optimizers and Where to Find Them
> > >
> > > [3] Sign-SGD via Parameter-Free Optimization, ICLR 2026
> > >
> > > [4] Muon: An Optimizer For Hidden Layers In Neural Networks
> > >
> > > [5] The Road Less Scheduled, NeurIPS
> > >
> > > [6] Prodigy: An Expeditiously Adaptive Parameter-Free Learner, ICML

---

### Official Review · Reviewer_z4Xn · 2026-03-13

**Soundness:** 3
**Presentation:** 2
**Significance:** 2
**Originality:** 3
**Overall Recommendation:** 5
**Confidence:** 3

**Summary:**

This work introduced a new algorithm called softsignum, to enable a smooth transition from sign-based update to clipped gradient update, mitigating issues such as learning rate mismatch and momentum buffer incompatibility. The benefit of the proposed algorithm is studied both theoretically and empirically, showing strong performance.

**Compliance With Llm Reviewing Policy:**

Affirmed.

**Final Justification:**

The authors provide a novel method for pretraining, providing both theoretical guarantees, as well as extensive numerical experiments. My initial judgement was positive and was further reinforced by the authors' rebuttal, which addressed the minor concerns I had. For these reasons, I recommend acceptance.

**Key Questions For Authors:**

Overall, I find the work interesting and appreciate the merits of the paper. At the same time, there are some shortcomings discussed above, which make it hard for me to give a higher score. I have one question to the authors: what if the sign-based updated is replaced by an adaptive algorithm, e.g., AdamW? Would you expect the algorithm to perform better?

**Limitations:**

Yes

**Strengths And Weaknesses:**

### Strengths

* The authors propose a novel algorithm and analyze its performance both theoretically and experimentally.

* Theoretical results show that the performance of softsignum under a constant temperature parameter can be tighter when gradient exhibits heterogeneity across iterations. While constant temperature is different from the approach proposed in Algorithms 3-4, it still sheds some light on an interesting point that the operator $\tanh(\cdot)$ could be useful in solving the real problem.

* The writing is clear and well-motivated.

* The proposed algorithm can preserve benefits such as memory efficiency, while simultaneously mitigating parameter heterogeneity and learning rate mismatch issues.

* A good amount of works has been devoted to validate the claims of this work. The benefit of softsignum has been well analyzed experimentally.

### Weaknesses

* Softsignum introduces many hyperparamaters need to be tuned carefully to ensure good performance. This limitation may hinder the applicability the proposed algorithm in a vast range of applications.

* Theoretical results do not properly reflect the convergence behavior of the proposed algorithm; in particular the proposed algorithm uses a time-varying temperature parameter and allows for different behavior during different training state.

* Adding a simulation showing how the temperature parameter $\tau$ changes would be helpful in understanding when the sign or clipping dominates over the other during the second training phase.

---

> ### Author Rebuttal · Authors · 2026-03-30
>
> We thank the reviewer for the positive assessment and the thoughtful questions. In response, we have substantially strengthened both the practical and theoretical components of our paper. Additional plots can be found here: https://anonymous.4open.science/r/softsignum-CA85/
>
> **(W1) Hyperparameters**
>
> First, we have updated our scheduling function to require only one hyperparameter. The scheduling now consists of two stages: sign and transition. The transition phase is now quantile-based: we assume that the components of the momentum follow a Cauchy distribution and, at the transition point, we estimate its parameters. Then, we use the quantiles of this distribution to determine the temperature such that, at each iteration T of the transition phase, a fraction 1 - T/T_trans of the components lies in the sign regime (results with new scheduling one can find in Q1). Thus, the only remaining parameter is $\alpha_{sign}$. Below, we demonstrate that SoftSignum is highly robust to it:
> | $\alpha_{sign}$ | 0.9 | 0.7 | 0.5 | 0.3 |
> |-|-|-|-|-|
> | SoftSignum (Next Token, Transformer) val loss | $1.413$ | $1.411$ | $1.408$ | $1.406$ |
> | SoftSignum (LLaMa), perplexity| $18.519$ | $18.530$ | $18.541$ | $18.559$ |
>
> **(W2) Theory**
>
> We incorporate the temperature parameter $\tau$ into the step (3) and convergence rate. In the submitted version, the generalized update (3) did not explicitly use $\tau$, and the convergence bound was stated in terms of $\delta$ alone. We revised the theory, now $\tau$ appears as an explicit parameter in the update and propagates through the entire analysis. Theorem 3.4 now reads: $$\frac{1}{K}\sum_{k=0}^{K-1} \frac{1}{\tau^2}\mathbb{E}[V^\star(-\tau\nabla f(\theta^k))] \leq \frac{f(\theta^0)-f^\star}{\delta\tau K} + (1-\beta)\sigma^2 + \frac{||\nabla f(\theta^0)||_2^2}{K(1-\beta)},$$
>
> under the condition $\delta\tau \leq \frac{1}{2L}\min \left(1, \frac{1-\beta}{\beta}\right)$. This formulation makes explicit that $\tau$ plays a role symmetric to $\delta$ in controlling convergence: the effective stepsize is $\delta\tau$. This provides theoretical insight into why scheduling $\tau$ is desirable: it is well-established that learning rate scheduling is profitable for training neural networks, and the key advantage of temperature formulation is that it allows for parameter-wise learning rate scheduling.
>
> **(W3) $\tau$ scheduling**
>
> Please see tau_scheduling.pdf in the anonymous repository for a visualization demonstrating how the sign and clipping regimes interact during training.
>
> **(Q1) Other methods**
>
> Our work is focused on sign-based methods and uniform sign approximation; thus, adaptive methods such as AdamW are out of our scope. However, the recently proposed Muon method, which outperforms AdamW in many DL scenarios, is fundamentally a sign-based method in spectral space, since $U sign(\Sigma) V^T = UV^T$. Thus, it suffers from the same terminal-phase weaknesses (1 and 2), and we therefore apply the same smoothing idea to its update. To ensure implementation efficiency via Newton Schultz iterations [1] for step calculation we replace tanh() with it's algebraic variant (i.e., approximation without exponents): $f(x) = \tau x/\sqrt{1+(\tau x)^2}$ (where $\tau$ is scheduled temperature as before). This function satisfies our theoretical assumptions, therefore the convergence guarantees of Theorem 3.4 carry over without modification, further demonstrating the generality of the proposed framework.
>
> We call this matrix modification SoftMuon. It outperforms the base Muon method in our next-token prediction setup and LLaMa pretraining ($\alpha_{sign}=0.9$, results averaged for 3 runs). This highlights that our approach improves even SOTA sign-based methods.
>
> | Method | SoftMuon | Muon | SoftSignum | Signum | AdamW |
> |-|-|-|-|-|-|
> | Next Token Prediction, Transformer, val loss | $\mathbf{1.389 \pm 0.004}$ | $1.406 \pm 0.006$ | $\mathbf{1.410 \pm 0.005}$  | $1.434 \pm 0.003$ | $1.421 \pm 0.006$ |
> | LLaMa pretraining, perplexity| $\mathbf{18.288 \pm 0.002}$ | $18.591 \pm 0.002$  | $\mathbf{18.519 \pm 0.002}$ | $18.688 \pm 0.002$ | $18.709 \pm 0.002$ |
>
> **Note that SoftMuon provides best performance, while SoftSignum is on par with Muon and outperforms AdamW and Signum.** Learning curves one can find in llama.pdf in the anonymous repo.
>
> In terms of theory, the hardest part of the analysis is the generalized descent lemma and momentum error bound, and it was already established in the submitted version. The matrix extension follows cleanly from the spectral theory of convex matrix functions [2]: for a matrix $D = U\Sigma V^T$, we define $f(D) = Uf(\Sigma)V^T$, where $f$ is applied element-wise to the singular values. The theoretical analysis requires no fundamental change in proof structure.
>
> [1] Jiang X., Semenov A., Stich S. U. Enhancing LLM Training via Spectral Clipping //arXiv preprint arXiv:2603.14315. – 2026.
>
> [2] Lewis A. S. Convex analysis on the Hermitian matrices

---

> > ### Author Rebuttal · Reviewer_z4Xn · 2026-04-01
> >
> > I thank the authors for their responses. My concerns have been addressed well and I find the paper has some good merits. Therefore, I am increasing my score accordingly.

---

### Decision · Program_Chairs · 2026-04-30

**Decision:**

Accept (regular)

**Comment:**

The paper proposes SoftSignum, an optimizer that uses a temperature-scheduled tanh transformation to smoothly transition from sign-based updates to clipped gradient updates. It contains convergence analysis and experiments across image classification, GNN training, and LLM pretraining (although reviewers disagreed about the validity of the relevance of the LLM experiments due to scale). Reviewers were split down the middle on this submission.

Reviewer z4Xn and Reviewer vejZ both recommended acceptance, citing the novelty (despite there being similar methods in the literature), the theoretical analysis, and the wide experimental evaluation. Reviewer z4Xn noted that the theoretical results provide useful insight into the benefits of the operator under gradient heterogeneity, and Reviewer vejZ (confidence 4) found the mathematical analysis sound and the experimental results convincing. The theoretical aspects constitute a major part of this work and were positively received.

The negative reviewers raised two main concerns. First, Reviewers GTw2 and 5dLW both argued that comparisons against AdamW, Lion, and Muon are missing from the LLM experiments, making it hard to assess practical competitiveness. The authors provided additional results during the rebuttal (including 720M-scale runs), though Reviewer GTw2 remained skeptical of these and Reviewer 5dLW found the rebuttal changes substantial relative to the original submission. Second, Reviewer 5dLW questioned the originality, noting that smooth transitions between Adam and SGD have been explored before (e.g., "A decreasing scaling transition scheme from Adam to SGD"), a reference not discussed in the paper.

These are legitimate concerns, but I do not find them sufficient to override the positive contributions of this work. The missing baseline issue was addressed as the authors did provide additional comparisons + sweeps during rebuttal. On originality, the specific mechanism (parameterized tanh with temperature scheduling applied per-parameter) is distinct from prior Adam-to-SGD, although discussion of related work on clipping methods should be improved (e.g., [1]).

Because of the split in opinions, there was a post-rebuttal discussion in which Reviewer 5dLW agreed that the paper has some merit despite its shortcomings in terms of evaluating this method for pretraining (the scales of the models are not in the Billion parameter regime). I don't disagree that this level of empirical validation is missing, but I also don't think it's absolutely necessary if the paper offers other contributions (meaningful small scale experiments, theoretical results). I also note that although Reviewer vejZ did not write a final justification, they did participate in the post-rebuttal discussion so I did not downweight their review during my decision making process.

I therefore recommend acceptance and ask the authors to include the additional baseline comparisons (AdamW, Lion, Muon) from the rebuttal and discuss the relationship to prior switching strategies + other clipping variants in the camera-ready version.

---

1. Pethick, Thomas, Wanyun Xie, Mete Erdogan, Kimon Antonakopoulos, Tony Silveti-Falls, and Volkan Cevher. "Generalized Gradient Norm Clipping & Non-Euclidean (L 0, L 1)-Smoothness." In NeurIPS 2025-39th Conference on Neural Information Processing Systems. 2025.